# Dual impact of elevated temperature on plant defence and bacterial virulence in *Arabidopsis*

Bethany Huot [1,2,3], Christian Danve M. Castroverde [1,3], André C. Velásquez[1], Emily Hubbard[1], Jane A. Pulman[4,5], Jian Yao[6], Kevin L. Childs [4,5], Kenichi Tsuda [7], Beronda L. Montgomery[1,2,8,9] & Sheng Yang He[1,2,3,4,9,10]

Environmental conditions profoundly affect plant disease development; however, the underlying molecular bases are not well understood. Here we show that elevated temperature significantly increases the susceptibility of *Arabidopsis* to *Pseudomonas syringae* pv. *tomato* (*Pst*) DC3000 independently of the phyB/PIF thermosensing pathway. Instead, elevated temperature promotes translocation of bacterial effector proteins into plant cells and causes a loss of ICS1-mediated salicylic acid (SA) biosynthesis. Global transcriptome analysis reveals a major temperature-sensitive node of SA signalling, impacting ~60% of benzothiadiazole (BTH)-regulated genes, including *ICS1* and the canonical SA marker gene, *PR1*. Remarkably, BTH can effectively protect *Arabidopsis* against *Pst* DC3000 infection at elevated temperature despite the lack of *ICS1* and *PR1* expression. Our results highlight the broad impact of a major climate condition on the enigmatic molecular interplay between temperature, SA defence and function of a central bacterial virulence system in the context of a widely studied susceptible plant–pathogen interaction.

[1] Department of Energy Plant Research Laboratory, Michigan State University, East Lansing, MI 48824, USA. [2] Cell and Molecular Biology Program, Michigan State University, East Lansing, MI 48824, USA. [3] Plant Resilience Institute, Michigan State University, East Lansing, MI 48824, USA. [4] Department of Plant Biology, Michigan State University, East Lansing, MI 48824, USA. [5] Center for Genomics Enabled Plant Science, Michigan State University, East Lansing, MI 48824, USA. [6] Department of Biological Sciences, Western Michigan University, Kalamazoo, MI 49008, USA. [7] Department of Plant-Microbe Interactions, Max Planck Institute for Plant Breeding Research, 50829 Cologne, Germany. [8] Department of Biochemistry and Molecular Biology, Michigan State University, East Lansing, MI 48824, USA. [9] Department of Microbiology and Molecular Genetics, Michigan State University, East Lansing, MI 48824, USA. [10] Howard Hughes Medical Institute, Michigan State University, East Lansing, MI 48933, USA. Correspondence and requests for materials should be addressed to B.L.M. (email: montg133@msu.edu) or to S.Y.H. (email: hes@msu.edu)

Plant diseases represent one of the most important causes of crop loss worldwide[1]; therefore, understanding the mechanisms underlying disease development is critical for developing effective disease control measures as part of global efforts to enable crop yields commensurate with increasing demand[1,2]. Weather plays a large role in determining the outcome of plant–pathogen interactions, and it has been noted that disease epidemics are more likely to occur when environmental conditions are suboptimal for the plant[3,4]. Responding to combined stresses (e.g., abiotic plus biotic) is challenging for plants because the response needed to mitigate one stress often can exacerbate another[4,5]. Breeding efforts to enhance yield typically reduce genetic diversity, which increases vulnerability to disease, and is also likely to negatively impact the resilience of plant immunity under adverse environmental conditions[1,6]. Increasing our understanding of how specific environmental factors affect the host and the pathogen as well as their interactions can inform strategies for developing robust crop resistance under increasingly unpredictable climate conditions.

The ability of the bacterial pathogen *Pseudomonas syringae* pv. *tomato* DC3000 (*Pst* DC3000) to cause disease in *Arabidopsis thaliana* (hereafter *Arabidopsis*) has made it a popular model for studying plant–pathogen interactions[7]. Two important virulence mechanisms employed by *Pst* DC3000 to cause disease are the phytotoxin coronatine (COR) and the type III secretion system (T3SS), which translocates bacterial effectors into host cells[7]. Conflicting results exist with respect to the effect of temperature on these virulence mechanisms. Elevated temperature has a negative effect on the expression of both COR-related and T3SS-related genes in vitro[8]. However, elevated temperature did not affect *Pst* DC3000 production of COR in planta[9]. Whether elevated temperature affects the production and translocation of type III bacterial effectors into plants is unknown.

Salicylic acid (SA) is a major plant defence hormone important for both local and systemic resistance against biotrophic and hemi-biotrophic pathogens, such as *Pst* DC3000[10]. In *Arabidopsis*, pathogen-induction of SA biosynthesis occurs predominantly through the isochorismate pathway involving ICS1[11]. Following SA induction, the master regulator, NPR1, accumulates in the nucleus where it interacts with TGA and WRKY transcription factors (TFs) to promote transcriptional reprogramming[12]. Among the many genes induced by SA, *PR1* is one of the most widely used markers for SA signalling in *Arabidopsis*[13]. SA-mediated defence has been well established as a crop protecting mechanism; for example, an SA synthetic analogue, benzothiadiazole (BTH), can provide resistance resulting in increased yield in multiple crops, including wheat[14] and maize[15]. Additionally, over-expression of *NPR1* has been shown to improve disease resistance in rice[16]. Basal defence against *Pst* DC3000 and induction of SA during effector-triggered immunity have been shown to be compromised at elevated temperature[17,18]. However, it is unclear whether either of these outcomes results from a direct impact of temperature on the SA pathway, as SA-deficient mutants were reported to retain temperature sensitivity during basal defence[18], and loss of effector-triggered immunity-induced SA may be an indirect effect resulting from temperature-mediated loss of upstream resistance (R) protein function. SA also plays a role in pattern-triggered immunity[19], which has been reported to not be suppressed at elevated temperature[20].

It was not clear until recently how plants sense elevated temperature. The phyB red-light photoreceptor is a negative regulator of the PIF4 growth-promoting TF[21], and was shown to function as a thermosensor in plants[22,23]. At elevated temperatures, heat inactivation of phyB results in de-repression of PIF4-regulated genes, enabling growth[22]. Another recent study suggests that PIF4 mediates defence suppression at elevated temperature;

however, all the *pif* mutants tested retained temperature-sensitive pathogen growth[24].

Although previous research has studied the effects of elevated temperatures on effector-triggered immunity or pattern-triggered immunity, our knowledge of the impact of elevated temperature on disease development—from pathogen virulence systems and host defence to temperature sensing—during a susceptible plant–pathogen interaction remains fragmented, preventing formulation of a cohesive model that could guide future research. We sought to address this knowledge gap by studying the model susceptible *Arabidopsis-Pst* DC3000 pathosystem. Contrary to prevailing results obtained in vitro, we discovered that elevated temperature has a positive effect on the function of the T3SS in planta, and a negative, but phyB/PIF-independent effect on SA-mediated defence in the host, resulting in overall enhanced disease. Remarkably, although BTH-induction of the temperature-sensitive *PR1/ICS1* branch is compromised at 30 °C, BTH-mediated protection against *Pst* DC3000 persists. We propose an integrated model illustrating the interplays between temperature, SA-mediated defence and the function of the T3SS in the context of a susceptible plant–pathogen interaction.

## Results

**Elevated temperature enhances disease susceptibility.** Temperatures between 27–30 °C are considered 'moderately elevated' for *Arabidopsis*, which is more susceptible to *Pst* DC3000 infection in this temperature range[18,24–27]. However, some of these studies used plants that were grown at acclimated temperatures for long periods of time before disease assays[18,27], resulting in plants with dramatically different morphology, including exaggerated hypocotyl and petiole elongation, due to PIF4-induction of auxin[28]. Although periods of elevated temperature vary in nature, to minimize the impact of physiological differences confounding our study, we first assessed the effect of a shorter, 2-day temperature acclimation period on infection of *Arabidopsis* plants by *Pst* DC3000. Four-week-old plants were acclimated to test chambers at 23 °C (control) or 30 °C (test) for only 48 h prior to syringe-infiltration with *Pst* DC3000. Morphological differences were greatly reduced between test and control plants acclimated for 48 h relative to those acclimated for 7 days (Supplementary Fig. 1a–c). We observed a 30-fold increase in bacterial growth as well as a dramatic increase in disease-associated chlorosis in plants at 30 °C relative to those at 23 °C (Fig. 1a, b). In contrast, *Pst* DC3000 grew similarly at 23 °C and 30 °C in vitro (Supplementary Fig. 2), suggesting that the effect of elevated temperature on *Pst* DC3000 multiplication is plant-dependent.

We next conducted *Pst* DC3000 infections in plants that were (i) acclimated and kept at 23 °C following infection (23 °C → 23 °C), (ii) acclimated at 30 °C and shifted to 23 °C (30 °C → 23 °C), (iii) acclimated at 23 °C and shifted to 30 °C following infection (23 °C → 30 °C) or (iv) acclimated and kept at 30 °C following infection (30 °C → 30 °C). We found that the post-infection temperature is the determining factor in disease susceptibility (Supplementary Fig. 3). Although plants were consistently more susceptible to *Pst* DC3000 at 30 °C relative to plants at 23 °C, the degree of enhanced disease susceptibility was more consistent following a 2-day acclimation at 30 °C (Supplementary Fig. 3). Therefore, we used a 2-day acclimation to 30 °C prior to inoculation for the remainder of our experiments.

PIF4 was recently proposed as a regulator of growth promotion and immunity suppression at elevated temperature[24]. To determine whether PIFs are responsible for enhanced susceptibility at elevated temperature in our system, disease assays were conducted with wild type (WT, Col-0) and *pif1 pif3 pif4 pif5*[29] (hereafter, *pifq*) mutant plants 1 dpi, 2 dpi and 3 dpi at both 23 °C and 30 °C. No difference in bacterial growth was observed

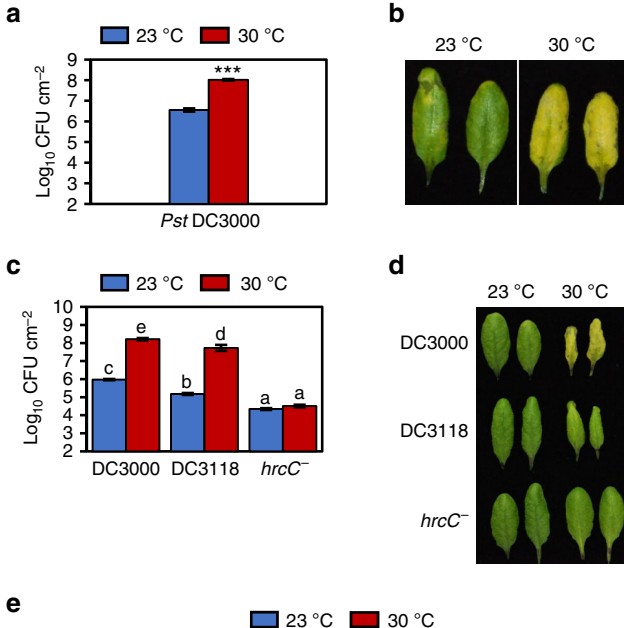

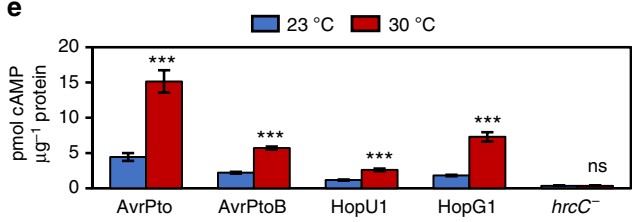

**Fig. 1** Enhanced growth of *Pst* DC3000 in planta at 30 °C requires the type III secretion system and results in elevated levels of effector translocation. **a** Bacterial growth in plants ($n = 4$) 3 days after syringe-infiltration with *Pst* DC3000. **b** Disease symptoms at three dpi for plants in (**a**). **c** Bacterial growth in plants ($n = 4$) 3 days after syringe-infiltration with *Pst* DC3000, *Pst* DC3118 (coronatine-deficient mutant) or *hrcC*⁻ (T3SS-deficient mutant) strains. **d** Disease symptoms three dpi for plants in (**c**). **e** Amount of cyclic AMP (cAMP) generated in temperature-acclimated plants syringe-infiltrated with *Pst* DC3000($P_{nptII}$::*avrPto-CyaA*) ($n = 4$), *Pst* DC3000($P_{tac}$::*avrPtoB-CyaA*) ($n = 4$), *Pst* DC3000($P_{tac}$::*hopU1-CyaA*) ($n = 6$), *Pst* DC3000 ($P_{tac}$::*hopG1-CyaA*) ($n = 6$) or *hrcC*⁻($P_{nptII}$::*avrPto-CyaA*) ($n = 4$) strains. Tissue was collected at 4–6 hpi for quantification of cAMP, which was normalized by total protein. Higher levels of cAMP indicate more translocation of bacterial effectors. All data are representative of three independent experiments; graphical data are presented as the mean ± standard error of the mean (s.e.m.), with $n$ = biological replicates. Letters indicate statistical significance based on a two-factor ANOVA with Tukey's HSD post hoc analysis ($p < 0.05$); samples sharing letters are not significantly different. Asterisks indicate statistical significance based on a Student's $t$ test (***$p < 0.001$) of pairwise comparisons for each individual effector strain at 23 °C vs. 30 °C; 'ns' indicates no significance

between the WT and *pifq* mutant plants at 30 °C for each time point assessed (Supplementary Fig. 4a), indicating that PIFs do not play a major role in mediating elevated temperature-dependent enhancement of disease susceptibility to *Pst* DC3000 inside the infected leaves (i.e., when bacteria are infiltrated into the leaf apoplast). Next, we examined the possibility that enhanced susceptibility at elevated temperature is due to heat inactivation of the phyB photoreceptor, which is a positive regulator of SA-induced *PR1* gene expression and has been shown to be a thermosensor for photomorphogenesis[22,23,30]. To test this, we assessed disease susceptibility in temperature stable phyB transgenic lines ($35S$::*PHYB*^Y276H, referred to as YHB hereafter)[31,32]. We observed no difference in disease phenotype between YHB and Landsberg *erecta* (L*er*) WT plants at either

temperature (Supplementary Fig. 4b). Taken together, we conclude that enhanced susceptibility at elevated temperature is independent of the phyB/PIF pathway.

**Elevated temperature enhances bacterial type III secretion.** We next examined whether enhanced multiplication of *Pst* DC3000 in planta at 30 °C requires bacterial virulence factors. Specifically, growth of two bacterial mutant strains, *Pst* DC3000 *hrcC*⁻ (hereafter *hrcC*⁻, defective in the T3SS[33]) and *Pst* DC3118 (defective in COR production[34]), was compared with growth of *Pst* DC3000 in *Arabidopsis* plants kept at 23 °C or 30 °C. The *hrcC*⁻ mutant strain had no detectable increase in growth at 30 °C (Fig. 1c). In contrast, while disease-associated leaf chlorosis was greatly reduced in *Pst* DC3118-infected plants (Fig. 1d), growth of this strain was 400-fold higher at 30 °C than at 23 °C (Fig. 1c), indicating that enhanced growth of *Pst* DC3000 in planta at 30 °C requires a functional T3SS but not COR.

We found a requirement of the T3SS for enhanced disease at 30 °C surprising because previous in vitro studies had shown a negative effect of elevated temperature on the T3SS[8]. To resolve this unexpected dilemma, we directly examined T3SS-mediated bacterial effector translocation into plant cells at 30 °C using *Pst* DC3000 strains with either the AvrPto, AvrPtoB, HopU1 or HopG1 effectors fused to the CyaA reporter[35]. To ensure differences in effector translocation were not influenced by bacterial populations, plant samples were first selected based on having similar bacterial populations at the 4–6 h post-infiltration (hpi) time point used for the translocation assay (Supplementary Fig. 5a, b). A *hrcC*⁻ mutant strain carrying the $P_{nptII}$::*avrPto-CyaA* plasmid was used as a negative control for the translocation assay. We observed a significant increase (2–4-fold) in translocation of all tested effectors at 30 °C compared to that at 23 °C (Fig. 1e), suggesting that, contrary to the prevailing notion from in vitro studies, increased *Pst* DC3000 virulence at 30 °C is linked to increased translocation of bacterial effector proteins.

**The SA pathway is compromised at elevated temperature.** As SA-mediated defence plays a major role in protecting *Arabidopsis* against virulent *Pst* DC3000, we investigated the possibility that the SA pathway is compromised at 30 °C. We measured *ICS1* and *PR1* marker gene expression as well as SA metabolite levels 24 hpi with mock or *Pst* DC3000. We observed significant *Pst* DC3000-induction of both *ICS1* (7-fold) and *PR1* (60-fold) in plants kept at 23 °C, whereas neither gene was induced by the pathogen at 30 °C (Fig. 2a). Similarly, total SA was induced by *Pst* DC3000 to levels 7-fold higher than in mock-infiltrated plants at 23 °C, but no significant increase was detected at 30 °C (Fig. 2b). To determine whether loss of SA can account for enhanced susceptibility at elevated temperature, we tested bacterial growth at 1 dpi, 2 dpi and 3 dpi in both WT (Col-0) and *sid2-2* (hereafter, *ics1*) mutant plants. As previously reported[18], *ics1* plants were more susceptible to *Pst* DC3000 than WT plants at 23 °C, showing 70-fold more bacterial growth at 2 dpi, which is prior to the bacterial population reaching saturation (Fig. 2c). Under our experimental conditions, the *ics1* mutant showed only a slight (3-fold) increase in bacterial growth at 1 dpi, but showed similar bacterial loads at 2 dpi and 3 dpi at 23 °C vs. 30 °C (Fig. 2c). As the disease phenotype of the *ics1* mutant at 23 °C resembles that observed in WT plants at 30 °C, we speculated that loss of pathogen-induced SA in WT plants is likely a major mechanism for enhanced disease at elevated temperature.

To determine whether increased translocation at elevated temperature occurs in addition to or because of loss of SA, we quantified translocation of bacterial effectors in WT and *ics1* mutants at 23 °C and 30 °C. As before, bacterial populations were

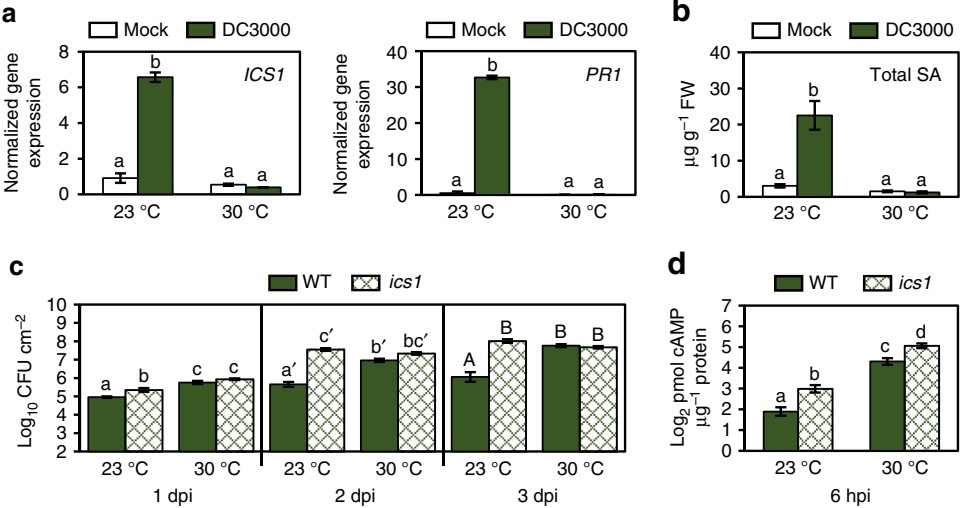

**Fig. 2** Enhanced disease at elevated temperature corresponds to loss of SA biosynthesis. **a** SA marker gene expression ($n = 3$) and (**b**) SA metabolite quantification ($n = 4$) 24 h after vacuum-infiltration with mock or *Pst* DC3000. qPCR was used for gene expression analysis, with expression of *ICS1* and *PR1* normalized to the expression of *PP2AA3*. SA and SAG metabolites were quantified using LCMS, converted to ng, normalized by sample fresh weight (FW) mass (g), and then combined and reported as total SA. **c** Bacterial growth in WT and *ics1* mutant plants ($n = 4$) at 1, 2 and 3 days after syringe-infiltration (dpi) with *Pst* DC3000. **d** Translocation of bacterial effectors in WT and *ics1* mutant plants ($n = 4$) syringe-infiltrated with *Pst* DC3000(P$_{nptII}$:: *avrPto-CyaA*). All data are representative of three independent experiments, and are presented as the mean ± s.e.m., with $n$ = biological replicates. Letters indicate statistical significance based on a two-factor ANOVA with Tukey's HSD post hoc analysis ($p < 0.05$); samples sharing letters are not significantly different. Data for each time point in (**c**) were analysed separately

assessed at the same time point as used for translocation assays to confirm differences in translocation are not due to differences in bacterial population (Supplementary Fig. 5c). We observed a significant increase (2-fold) in bacterial effector translocation into *ics1* vs. WT plants at 23 °C (Fig. 2d), indicating that loss of SA enables more effector translocation. However, there was also significantly more translocation of effector proteins into both WT and *ics1* mutant plants at 30 °C (4–5-fold, Fig. 2d), indicating that, at elevated temperature, both loss of SA biosynthesis in the host and increased bacterial translocation of effector proteins likely contribute to enhanced disease development.

**Suppression of SA responses at 30 °C is pathogen independent**. We next used BTH to directly interrogate the effect of elevated temperature on SA biosynthesis and signalling in a pathogen-free system. BTH is widely used as a surrogate for the SA signal, and is a potent inducer of SA response genes in *Arabidopsis*[10,36]. BTH significantly induced both *ICS1* (10-fold) and *PR1* (>2500-fold) gene expression as well as total SA levels (15-fold) in plants kept at 23 °C with no significant induction of *ICS1*, *PR1* or SA levels observed at 30 °C (Fig. 3a, b). Similarly, BTH-induction of callose deposition, which is thought to reinforce plant cell walls against pathogen penetration[37], was observed at 23 °C (8-fold) but not at 30 °C (Fig. 3c, d). In contrast, although slightly reduced at elevated temperature, flg22 elicited a strong callose response at both 23 °C (90-fold) and 30 °C (50-fold; Supplementary Fig. 6a, b), indicating that the observed effect of temperature on BTH-induced callose deposition is likely due to compromised SA signalling rather than an effect on the callose synthase enzyme.

**NPR1 nuclear localization is retained at 30 °C**. Elevated temperature negatively affects R protein-mediated disease resistance[19,20]. Zhu et al. (2010) showed that loss of R protein nuclear localization contributes to compromised R-mediated defence at elevated temperature (28 °C)[38]. NPR1 is a key regulator

of SA signalling and accumulates in the nucleus upon SA signal perception[10,12]. As nuclear localization of NPR1 is required for *PR1* gene induction[39], it is possible that nuclear exclusion of NPR1 results in loss of *PR1* gene induction at 30 °C. To examine this possibility, we generated transgenic lines expressing a functional NPR1 protein tagged at the C-terminal end with the yellow fluorescent protein (YFP) under control of the native NPR1 promoter (*pNPR1::NPR1-YFP*, hereafter, *NPR1-Y1*; Supplementary Fig. 7b–d) in a confirmed *npr1* knock-out mutant (Supplementary Fig. 7a, c, d). Transgenic lines expressing the YFP protein under the control of the constitutive *35S* promoter (*p35S:: YFP*) were also generated as controls. A nuclear YFP signal was observed in mock-treated and BTH-treated *NPR1-Y1* plants at both 23 °C and 30 °C, although the signal was extremely weak in mock-treated plants at both temperatures (Fig. 4a). There was no observable effect of treatment (chemical or temperature) on YFP signal detected in the *p35S::YFP* control, and no YFP signal was detected in the parent *npr1* plants (Supplementary Fig. 8a, b).

To independently confirm NPR1 nuclear localization, subcellular fractionation experiments were also conducted. NPR1-YFP was observed in whole-cell lysate and both fractions of BTH-treated samples at both temperatures (Fig. 4b). Western blotting of UGPase and H3 proteins, which were used as cytosolic-fraction and nuclear-fraction specific markers, respectively, showed significant enrichment within their respective fractions (Fig. 4b). Thus, the confocal microscopic and nuclear fractionation data both show that elevated temperature does not prevent BTH-induced nuclear accumulation of the NPR1 protein.

To confirm the loss of BTH-induction of PR1 protein accumulation, western blot analysis using a PR1 antibody was also conducted with fractionated protein samples confirmed to have NPR1 localized to the nucleus. Similar to *PR1* gene expression (Fig. 3a), PR1 protein was only detectable in BTH-treated plants at 23 °C (Supplementary Fig. 9b). Therefore, in addition to loss of SA biosynthesis, there appears to be a negative effect of elevated temperature on SA-mediated signalling

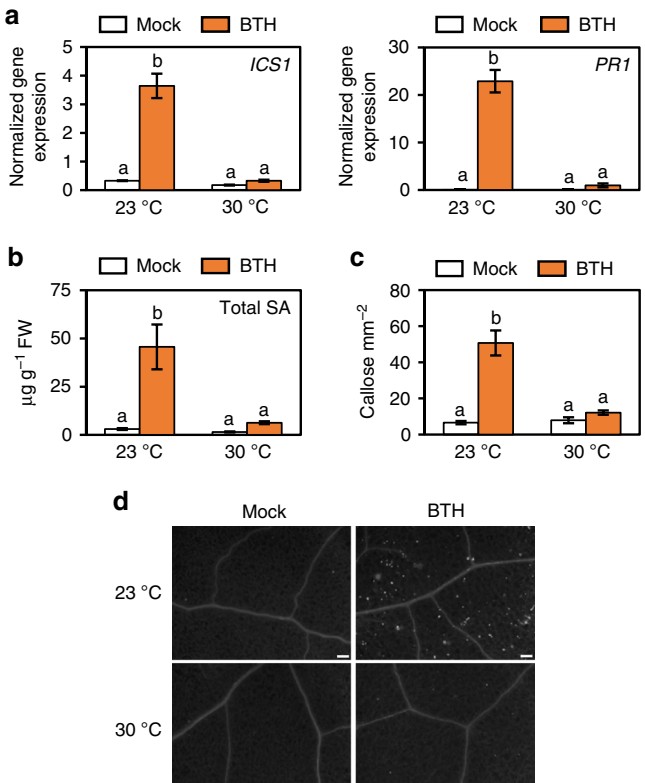

**Fig. 3** Induction of SA defence responses is compromised at 30 °C. **a** SA marker gene expression in plants ($n = 6$) 24 h after spraying with mock or BTH. **b** SA metabolite quantification in plants ($n = 4$) 48 h after spraying with mock or BTH. Gene expression and LCMS data were processed and analysed as described in Fig. 2. **c** Quantification and **d** representative images of callose accumulation from plants ($n = 6$) 24 hpi with mock or BTH. Scale bar length represents 100 μm. All data are representative of three independent experiments. All graphical data are presented as the mean ± s.e.m., with $n =$ biological replicates. Letters indicate statistical significance based on a two-factor ANOVA with Tukey's HSD post hoc analysis ($p < 0.05$); samples sharing letters are not significantly different

downstream of NPR1 leading to *PR1* expression and protein accumulation.

**Temperature bifurcates the SA-regulated transcriptome.** The dramatic effect of elevated temperature on BTH-induction of the canonical SA marker genes *PR1* and *ICS1* prompted us to conduct RNA sequencing (RNA-seq) to determine the extent of elevated temperature's impact on BTH-mediated global transcriptional reprogramming. We identified 169 differentially expressed genes (DEGs) in response to temperature alone (23 °C mock vs. 30 °C mock) and 2820 DEGs in response to BTH (BTH vs. mock at 23 °C or 30 °C) using cut-off criteria of 4-fold change in expression ($\log_2$ value $\geq \pm 2$) and Storey's $q$-value $< 0.01$ ($q$-values calculated from $p$-values based on $t$ test, see Supplementary Note 1 for further details; Supplementary Data 1). Based on gene ontology (GO) analysis of the 23 °C mock vs. 30 °C mock DEGs, genes upregulated at 30 °C are primarily involved in response to abiotic stress, such as heat, whereas genes downregulated at 30 °C are primarily annotated as being involved in responses to biotic stress (Supplementary Table 1, Supplementary Data 2).

Because changes in basal gene expression due to temperature will impact fold change, we analysed the BTH vs. mock DEGs in two ways: (i) fold change (BTH/mock) and (ii) expression levels (23 °C BTH vs. 30 °C BTH). k-means cluster analysis ($k = 9$) was

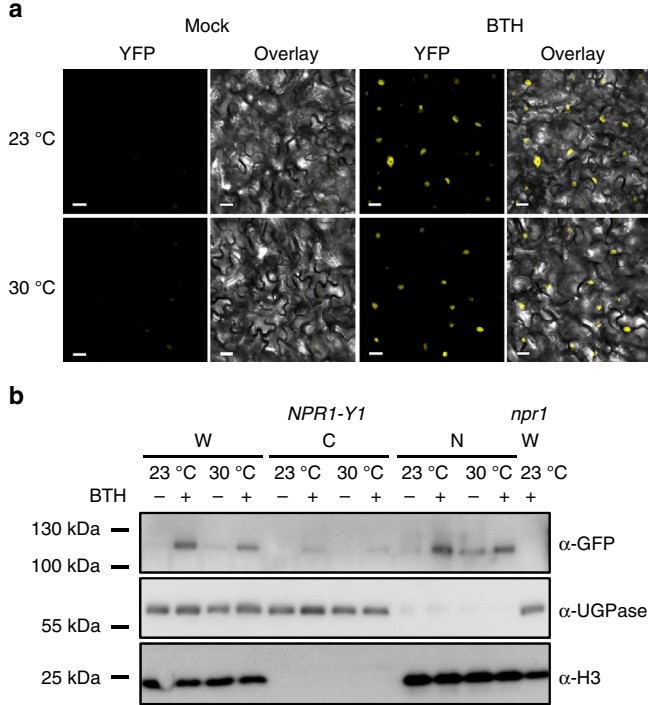

**Fig. 4** Induction of NPR1 nuclear localization is retained at 30 °C. **a** Representative confocal microscopy images of *NPR1-YFP* plants ($n = 4$) 24 h after spraying with mock or BTH. Images are of YFP (yellow) alone or YFP overlaid on Brightfield (BF, grey-scale). Scale bar length represents 10 μm. **b** Western blots of whole-cell lysate (W), non-nuclear (C, cytosolic) and nuclear (N) enriched fractions isolated from *NPR1-Y1* transgenic plants treated with mock (−) or BTH (+) solutions at 23 °C or 30 °C. Equal volumes (10 μl) of each protein sample were loaded and run in two separate 4–12 % gradient SDS-PAGE gels. Following transfer to PVDF membranes, one blot was probed with α-GFP primary antibody to detect the NPR1-YFP protein while the other blot was cut in two and the upper portion probed with α-UGPase (a cytosolic protein control) and the lower portion probed with α-H3 (a nuclear protein control). Whole-cell lysate extracted from *npr1* plants treated with BTH at 23 °C was used as the negative control for the NPR1-YFP band. All data are representative of three independent experiments

conducted with the 2820 BTH-responsive DEGs, and a heat map was generated to visualize gene expression patterns. Clusters 1–7 contain genes induced by BTH (1833 genes), with cluster 1, 2 and 3 genes exhibiting compromised induction, cluster 4 genes having similar induction, and cluster 5, 6 and 7 genes showing enhanced induction at 30 °C; clusters 8 and 9 contain genes suppressed by BTH at 23 °C but compromised in suppression at 30 °C (987 genes, Supplementary Fig. 10). For the analysis based on expression levels, a 2-fold cut-off was used to categorize genes as follows: group A, expression lower at 30 °C vs. 23 °C (956 genes); group B, expression similar at 30 °C and 23 °C (725 genes); and group C, expression higher at 30 °C vs. 23 °C (152 genes; Fig. 5a). As 810 out of 987 BTH-suppressed genes showed higher expression at 30 °C, these genes were kept in a single group D (Fig. 5a). A comparison of gene distribution by cluster and group reveals that 42% of genes that did not show differential induction based on cluster analysis (i.e. cluster 4) showed compromised expression at 30 °C based on group analysis (i.e. group A), indicating that cluster-based and group-based analyses have different sensitivities (Supplementary Table 2). Based on meta-analysis of publicly available microarray datasets (clusters only) and functional annotations using GO analyses, BTH-induced genes (clusters 1–7 / groups A–C) are involved in

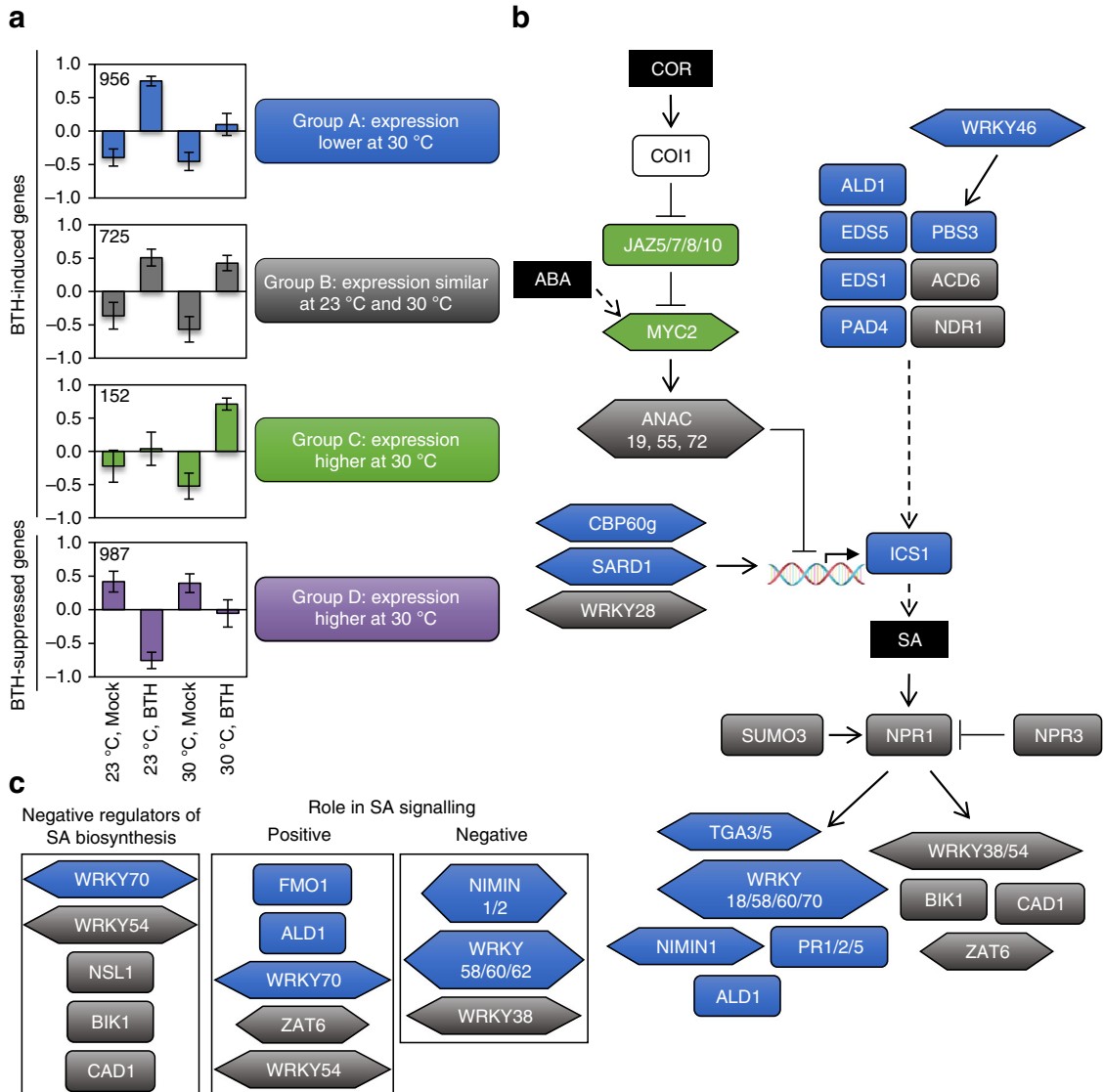

**Fig. 5** Global transcriptome analysis reveals a temperature-sensitive bifurcation in the SA signalling network. **a** RNA-seq gene expression profiles. Tissue was harvested for RNA extraction and sequencing 24 h after spraying plants with mock or BTH solutions. Differentially expressed genes (DEGs) and BTH-induced (groups A–C) and BTH-suppressed (group D) genes were determined as described in the text. Genes were grouped based on a 2-fold change difference in expression between BTH-treated samples at 30 °C vs. 23 °C. Data are represented as the mean ± s.e.m. of the centred, normalized expression values for each sample type (23 °C, Mock; 23 °C, BTH; 30 °C, Mock; 30 °C, BTH) within each group. The number of genes within each group is shown in the upper left corner of each graph. **b** and **c** Graphical depiction of genes involved in promotion or suppression of SA biosynthesis via regulation of the *ICS1* gene as well as genes involved in the SA signalling pathway. Genes are colour-coded based on the gene group **a** to which they belong. Genes not differentially regulated within the RNA-seq dataset are depicted using shapes without fill. Metabolites are denoted by black boxes with white text; transcription factors are denoted by a hexagon shape. The DNA molecule with bent arrow designates the *ICS1* gene promoter. Solid arrows show direct interactions, dotted arrows show indirect interactions and bar-headed lines show inhibitory interactions. References for each gene function depicted is provided in Supplementary Table 4. NPR1-regulated genes in **b** determined based on microarray study conducted by Wang et al[70]

various biotic or abiotic stress responses, whereas BTH-suppressed genes (clusters 8 and 9 / group D) are involved in photosynthesis and growth-related processes (Supplementary Figs. 11–13, Supplementary Table 3, Supplementary Data 3, 4, and 6).

If we consider genes induced by BTH at 23 °C (groups A and B, 1681 genes) as the typical SA response genes, a clear temperature-sensitive bifurcation is revealed at the 24 h time point assessed in our study, with 57% of BTH-induced genes 'temperature-sensitive' (group A, 956 of 1681 genes) and 43% of BTH-induced genes 'temperature-insensitive' (group B, 725 of 1681 genes). Based on this analysis, known positive regulators of SA biosynthesis, including *EDS1*, *PAD4*, *CBP60g* and *SARD1*[10],

are predominantly in the *PR1/ICS1* branch (group A), while negative regulators, such as *WRKY54*, *MYC2* and several MYC2-regulated *ANACs*[10], are in groups B and C (Fig. 5b, c, Supplementary Table 4). *NPR1* and two of its known regulators, *NPR3*[10] and *SUMO3*[40], are in group B (Fig. 5b), which is consistent with our finding that NPR1 induction (and localization) is not impaired at 30 °C. Downstream of NPR1, many SAR-related genes, including *PR1*, *PR2*, *PR5*, *ALD1* and *FMO1*[10], appear in group A; however, several positive regulators, including *WRKY54*[10] and *ZAT6*[41] are in group B (Fig. 5b, c).

We next conducted analysis of motif enrichment (AME)[42], using 1000 bp upstream of each gene, to identify TFs that may be involved in regulation of genes within each cluster and

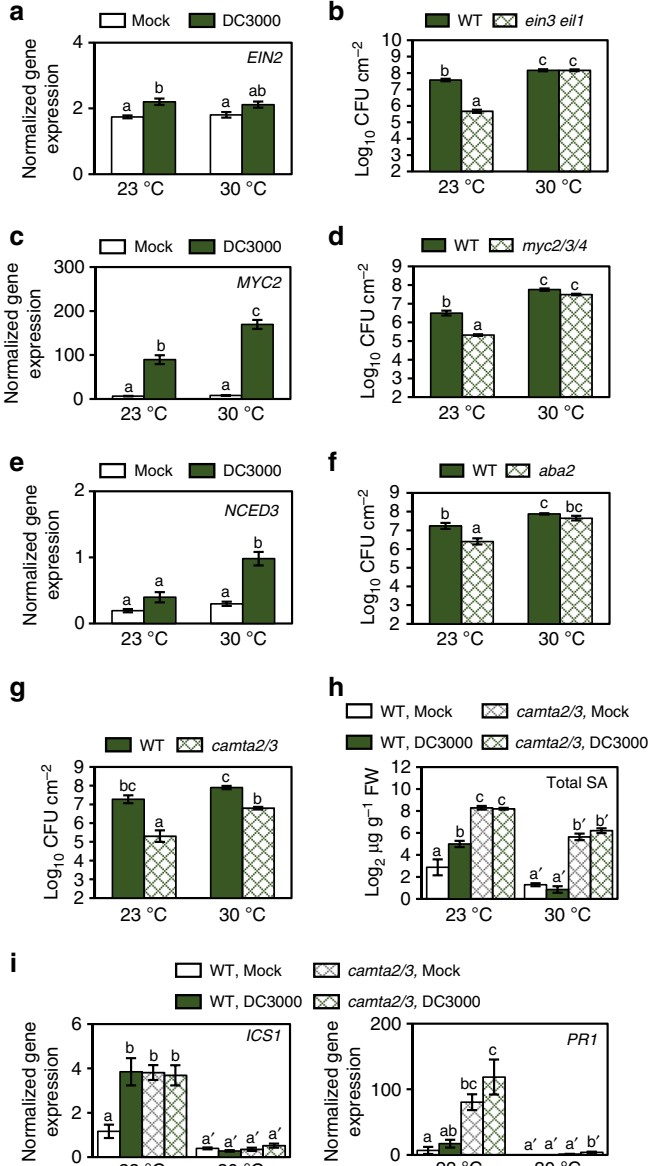

**Fig. 6** Major SA antagonistic pathways are not responsible for enhanced susceptibility to *Pst* DC3000 at elevated temperature. **a** ET, **c** JA and **e** ABA marker gene expression in plants (*n* = 3) 24 h after vacuum-infiltration with mock or *Pst* DC3000. Bacterial growth in WT and **b** *ein3 eil1*, **d** *myc2/3/4*, **f** *aba2* and **g** *camta2/3* mutant plants (*n* = 4) 3 days after vacuum-infiltration with *Pst* DC3000. **h** SA metabolite quantification and **i** SA marker gene expression in WT and *camta2/3* mutant plants (*n* = 4) 3 days after vacuum-infiltration with mock or *Pst* DC3000. Gene expression and LCMS data were processed and analysed as described in Fig. 2. Data are presented as the mean ± s.e.m. with *n* = biological replicates and are representative of three independent experiments. Letters indicate statistical significance based on a two-factor ANOVA with Tukey's HSD post hoc analysis (*p* < 0.05); samples sharing letters are not significantly different. Data in **h** and **i** were analysed in two groups based on temperature

group (Supplementary Data 5). The cluster-based analyses were difficult to interpret, as the same *cis* elements were identified as over-represented in several clusters (Supplementary Table 5a); however, the group-based analyses were more informative. While genes in groups A and B had an over-representation of the W-box motif bound by WRKY TFs, known to regulate genes induced by both biotic and abiotic stresses[43], only genes in group A had an over-representation of the *as-1* element bound by TGA TFs[44],

which are known to regulate SA-responsive *PR* genes[10] (Supplementary Table 5b). Genes containing *cis* elements bound by TF regulating circadian responses, such as CCA1[45], were over-represented in group B, which is the temperature-insensitive group. Additionally, genes in group C had an over-representation of *cis* elements bound by TFs involved in jasmonic acid (JA) and abscisic acid (ABA) signalling as well as the CAMTA2 and CAMTA3 TFs, both of which are negative regulators of SA[46,47]. Finally, group D genes had an over-representation of *cis* elements bound by PIFs, MYCs and multiple TFs involved in growth-related processes.

**Enhanced susceptibility is not due to major SA antagonists.**
Our GO and promoter analyses showed that plants treated with BTH at 30 °C have an upregulation of genes involved in JA, ethylene (ET) and ABA signalling pathways (Supplementary Tables 3 and 5), which are known to antagonize SA biosynthesis and signalling[48–50]. We conducted further experiments to determine whether these pathways are also upregulated during *Pst* DC3000 infection at elevated temperature. We found no measurable effect of temperature on pathogen-induction of ET-responsive genes, *EIN2* and *ERF6*, following infection; however, both JA pathway marker genes, *LOX2* and *MYC2*, as well as the *NCED3* gene involved in ABA biosynthesis were induced to higher levels by *Pst* DC3000 at 30 °C compared to those at 23 °C (Fig. 6a, c, e; Supplementary Fig. 14a, c). ABA metabolite levels were also increased following pathogen infection at 30 °C (Supplementary Fig. 14e). Therefore, while SA biosynthesis and signalling are compromised, *Pst* DC3000-induction of JA-mediated signalling and ABA biosynthesis are enhanced at elevated temperature.

To test whether ABA, JA or ET has a causal role in enhanced susceptibility at elevated temperature, we conducted disease assays at 23 °C and 30 °C using the *aba2-1*[51,52] (hereafter *aba2*) ABA-deficient mutant, the *dde2-2*[53] JA-deficient mutant (defective in the *ALLENE OXIDE SYNTHASE* gene; hereafter *aos*), the *myc2 myc3 myc4*[54] (hereafter *myc2/3/4*) JA-signalling mutant and the *ein2-1*[55] (hereafter *ein2*) and *ein3 eil1*[48] ET-signalling mutants. As expected, all five mutants showed a significant (5–10-fold) reduction of *Pst* DC3000 growth at 23 °C compared to WT plants (Fig. 6b, d, f; Supplementary Fig. 14b, d). However, based on the retention of temperature sensitivity in the mutant plants and similar levels of susceptibility between these mutants and the corresponding WT plants at 30 °C, we conclude none of these hormone signalling pathways is primarily responsible for enhanced susceptibility to *Pst* DC3000 at elevated temperature.

We also found enrichment of CAMTA-targeted *cis* elements in genes more highly induced and/or expressed following BTH treatment at 30 °C (Supplementary Table 5). CAMTA1, CAMTA2 and CAMTA3 are TFs that function redundantly in repressing SA biosynthesis at temperatures between 19 °C and 22 °C[47,56]. This repression is relieved upon pathogen perception or in response to cold temperatures (~4 °C), enabling SA accumulation in these conditions[56,57]. As expected, the *camta2 camta3* double mutant (hereafter *camta2/3*) showed enhanced resistance against *Pst* DC3000 relative to WT plants at 23 °C (90-fold less growth; Fig. 6g). Unlike the other mutants of SA negative regulators tested, the *camta2/3* mutant retained partially heightened resistance at 30 °C (13-fold less growth; Fig. 6g), indicating that CAMTAs may contribute to enhanced susceptibility at elevated temperature. In support of this, total SA in the *camta2/3* mutants was significantly elevated at 30 °C, with levels similar to pathogen-treated WT plants at 23 °C (Fig. 6h). However, although *ICS1* and *PR1* gene expression was constitutively elevated in the *camta2/3* mutant relative to WT plants at 23 °C, expression of these genes remained compromised at 30 °C (Fig. 6i), indicating that loss of *ICS1* and *PR1* gene expression at

elevated temperature is not due to CAMTA2/3-mediated suppression.

**BTH confers resistance at 30 °C without inducing *ICS1* or *PR1*.** BTH is an SA analogue available commercially as a means of crop disease control[58]. The dramatic effect on SA biosynthesis and signalling at 30 °C raises the possibility that BTH may fail to protect plants against pathogens at elevated temperature. To directly test this possibility, we conducted BTH protection assays in WT plants at 23 °C and 30 °C. To our surprise, BTH-treated WT plants had 150–200-fold less bacterial growth and showed no disease symptoms relative to the mock-treated controls at both temperatures (Fig. 7a, b). Even more surprising is that BTH protection occurred with no induction of the *ICS1* and *PR1* marker genes or accumulation of SA at 30 °C (Fig. 7c, d). Thus, BTH can fully protect *Arabidopsis* against *Pst* DC3000 in the absence of the canonical SA marker genes, *ICS1* and *PR1*.

We next tested whether BTH-mediated protection at 30 °C requires the core SA signalling pathway. BTH protection assays were conducted in WT, *npr1* mutant and *tga2 tga5 tga6*[59] (hereafter *tga2/5/6*) triple mutant plants. Mock-treated *npr1* mutant plants had 30-fold higher bacterial growth relative to mock-treated WT plants at 23 °C; BTH no longer provided protection in the *npr1* mutant plants at either temperature (Fig. 7e). Similarly, mock-treated *tga2/5/6* mutant plants had 10-fold more bacterial growth than mock-treated WT plants at 23 °C (Fig. 7f). BTH protection is completely absent in *tga2/5/6* mutants at 30 °C (Fig. 7f), indicating that NPR1, TGA2, TGA5 and TGA6 are required for BTH-mediated protection at elevated temperature.

Next, we tested the effect of BTH on translocation of bacterial effectors to determine whether BTH-mediated restriction of virulence proteins into plant cells contributes to protection at elevated temperature. A higher bacterial inoculum was used in BTH-treated plants than in mock-treated plants to enable equal bacterial populations in plants at 4 hpi (Supplementary Fig. 15a). We observed a significant reduction (2–3-fold) in translocation of AvrPto-CyaA effector protein in BTH-treated plants relative to the mock-treated controls at both temperatures (Fig. 7g). However, although the trends were consistent across all experiments, the effect size of BTH on bacterial effector translocation was more variable at 30 °C (Supplementary Fig. 15b). From these data, we conclude that BTH restriction of bacterial effector translocation contributes to protection at both temperatures, but there are likely other mechanisms involved in BTH-induced disease resistance.

## Discussion

According to the long-standing 'disease triangle' dogma in plant pathology, plant–pathogen interactions can only be fully understood within the context of environment[60]. Elevated temperature has been shown to break down effector-triggered immunity and promote disease in many plant-pathosystems[61], from fungal infection of wheat[62] to viral infection of tobacco[63]. While some insight regarding the molecular mechanisms involved in loss of effector-triggered immunity has been gained[64], the underlying cause for a potentially general enhancement of disease susceptibility in compatible plant–pathogen interactions (i.e. in the absence of effector-triggered immunity) was poorly understood prior to this study. We provide evidence that enhanced multiplication of virulent *Pst* DC3000 inside *Arabidopsis* leaves at elevated temperature is independent of the phyB/PIF thermosensing pathway (Supplementary Fig. 4a, b). Instead, enhanced bacterial multiplication at 30 °C requires a functional T3SS (Fig. 1c), and, contrary to the widely held notion based on studies of the T3SS *in vitro*[8], is associated with increased translocation of

bacterial effectors into plant cells (Fig. 1e). We also show that enhanced susceptibility of *Arabidopsis* under our conditions (i.e. when bacteria are pressure-infiltrated directly into the plant leaves) is linked to loss of SA, and the SA-deficient *ics1* mutant

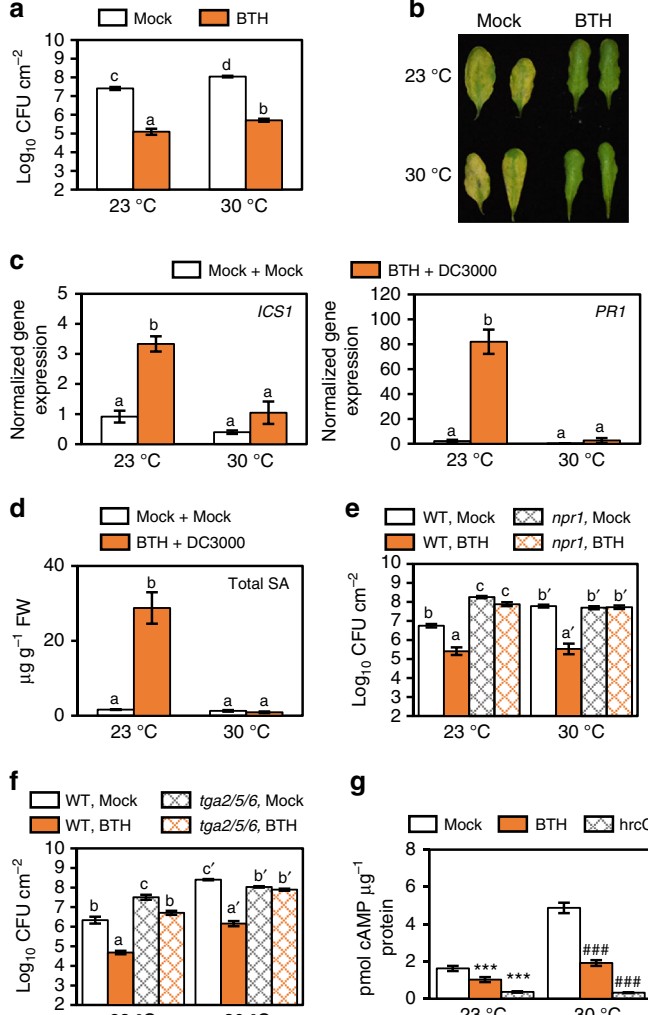

**Fig. 7** BTH protection against *Pst* DC3000 at 30 °C requires NPR1 and TGAs but occurs without *ICS1* and *PR1* expression. **a** Bacterial growth in mock- or BTH-pre-treated plants ($n = 4$) 3 days after vacuum-infiltration with *Pst* DC3000. **b** Disease symptoms three dpi in plants in **a**. **c** SA marker gene expression ($n = 3$) and **d** SA metabolite quantification ($n = 4$) 24 hpi of plants in **a**. Gene expression and LCMS data were processed and analysed as described in Fig. 2. Bacterial growth in mock- or BTH-pre-treated WT, **e** *npr1* and **f** *tga2/5/6* mutant plants ($n = 4$) 3 days after vacuum-infiltration with *Pst* DC3000. **g** Translocation of bacterial effector proteins in plants ($n = 4$) pre-treated with mock or BTH 24 h before syringe-infiltration with *Pst* DC3000($P_{nptII}$::avrPto-CyaA). Additional mock-treated plants ($n = 4$ for each temperature) were infiltrated with hrcC⁻($P_{nptII}$::avrPto-CyaA strains) as a negative control. Tissue was collected at 4 hpi for quantification of cAMP, which was normalized by total protein. All data are representative of three independent experiments; graphical data are presented as the mean ± s.e.m. with $n$ = biological replicates. Letters indicate statistical significance based on a two-factor ANOVA with Tukey's HSD post hoc analysis ($p < 0.05$); samples sharing letters are not significantly different. Data in **e** and **f** were analysed in two groups based on temperature as indicated by the prime symbol ('). Symbols in **g** denote statistical significance based on a one-factor ANOVA with Dunnett's post hoc analysis (***, ###$p < 0.001$) using the mock-treated sample at each temperature as the means for comparison

exhibits a largely temperature-insensitive disease phenotype, with a similar level of susceptibility as WT plants at 30 °C (Fig. 2c). RNA-seq analysis of combinations of temperature and BTH treatments, reveal temperature-sensitive and temperature-insensitive branches in the SA gene expression network (Fig. 5). We also provide evidence that BTH confers robust protection against disease at 30 °C even in the absence of canonical *ICS1* and *PR1* gene expression (Fig. 7a–c). Collectively, these results highlight the multi-faceted impact of elevated temperature on the molecular interplay between temperature, SA-mediated defence and the function of a central bacterial virulence system in one of the best studied susceptible plant–pathogen interactions.

We considered three potential models that could explain how elevated temperature affects the compatible host-pathogen interaction to enable enhanced disease. Based on previous in vitro analyses showing a negative effect of elevated temperature on the expression of virulence-associated genes[8] and our data showing loss of pathogen-induced SA biosynthesis (Fig. 2b), Model 1 predicts that elevated temperature negatively affects both pathogen virulence and SA-mediated defence. In this model, enhanced disease would result from increased host susceptibility in spite of compromised bacterial virulence. However, the inability of the non-pathogenic $hrcC^-$ mutant strain to grow more in plants at elevated temperature indicates that a functional T3SS is required for enhanced bacterial growth at 30 °C (Fig. 1c). Moreover, our results show that effector translocation in planta is more efficient at 30 °C (Fig. 1e), effectively refuting Model 1.

Alternatively, Model 2 proposes that elevated temperature enhances bacterial virulence, which then causes loss of SA biosynthesis to promote disease. *Pst* DC3000 generated COR, which is known to specifically target and antagonize SA biosynthesis by activating JA-mediated signalling[50], was previously shown to be unaffected by elevated temperature in planta[9]. Although symptom development was greatly reduced in plants infected with a COR-deficient mutant, growth of this strain was enhanced similarly to that of *Pst* DC3000 at 30 °C (Fig. 1c, d), thereby eliminating COR-mediated suppression of SA biosynthesis as a potential mechanism. As bacterial effector translocation is increased at 30 °C, it is possible that an effector-mediated process may contribute to loss of SA biosynthesis at elevated temperature. However, the loss of BTH-induced SA production at 30 °C (Fig. 3b), which is a pathogen-free treatment, prompts rejection of Model 2.

Therefore, we propose a third model whereby elevated temperature both antagonizes SA accumulation in the host and promotes translocation of virulence proteins by the pathogen resulting in enhanced disease (Fig. 8). Increased bacterial effector translocation into the *ics1* mutant relative to WT plants at 23 °C (Fig. 2d) suggests that loss of SA may result in increased 'permeability' of the cell to T3SS-mediated pathogenesis. This enhanced permeability would result in the observed temperature sensitivity of the *ics1* mutant to effector delivery. We typically observe a positive correlation between bacterial effector translocation and bacterial multiplication. However, at 30 °C, there is significantly more bacterial effector translocation into *ics1* mutant relative to WT plants with no further increase in bacterial populations (Fig. 2c, d). This suggests there is a minimum threshold of bacterial effector translocation required for maximal bacterial multiplication, after which no further positive effect may be observed.

One outstanding question to be addressed is the mechanism responsible for loss of SA at elevated temperature. Based on in vitro activity assays, which show the ICS1 enzyme maintains >90% maximal activity from 4 to 37 °C[65], and our data showing complete loss of *ICS1* gene induction at 30 °C, it seems likely that loss of SA biosynthesis and accumulation occurs upstream of *ICS1* gene expression. However, we cannot rule out the possibility of additional changes in ICS1 and other SA biosynthesis protein

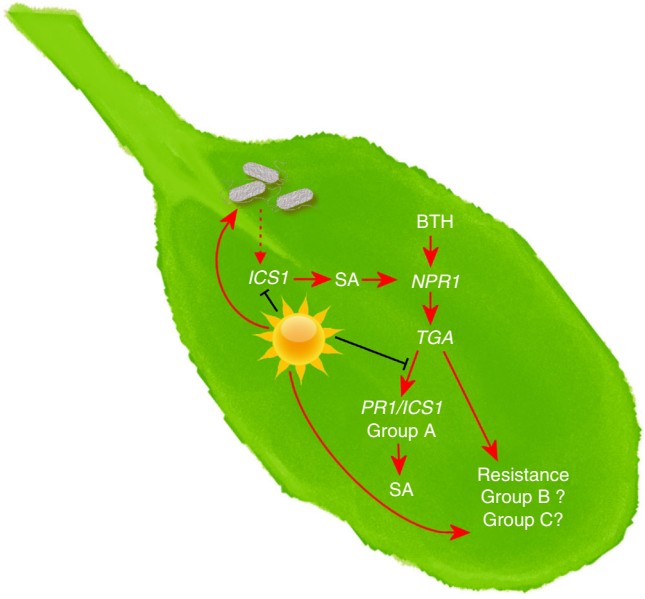

**Fig. 8** Model for the *Arabidopsis-Pst* DC3000 interaction at elevated temperature. At elevated temperature (depicted by the sun), pathogen-induction of SA biosynthesis via the ICS1 enzyme is blocked and translocation of bacterial T3E proteins is enhanced to promote disease. SA signalling induced by the SA synthetic analogue, BTH, is also affected, with the *PR1/ICS1* branch (group A in Fig. 5a) no longer induced. However, BTH-mediated resistance against *Pst* DC3000 is still conferred at elevated temperature in a NPR1- and TGA2/5/6-dependent manner. It is possible that genes in the temperature-insensitive branch, group B and/or genes in the elevated temperature + BTH-induced branch, group C, are involved in BTH-mediated resistance under this condition. Solid arrows represent direct positive interactions, dotted arrow represents induction of *ICS1* in response to pathogen detection and bar-headed lines represent inhibitory interactions

modifications at elevated temperature that may affect their activities. In this study, we discovered upregulation of a number of known negative regulators of ICS1-mediated SA biosynthesis, including components of JA-mediated[50] and ET-mediated[48] signalling and ABA[46] biosynthesis (Fig. 6a, c, e; Supplementary Fig. 14a, c, e). However, all hormone mutants tested showed similar levels of susceptibility as the WT plants at 30 °C (Fig. 6b, d, f; Supplementary Fig. 14b, d), indicating that none of these pathways is solely responsible for SA suppression at elevated temperature.

Retention of resistance in the *camta2/3* mutants and in BTH-treated WT plants at 30 °C in the absence of *PR1* and *ICS1* gene induction (Figs. 6g, i and 7a, c) is striking. In particular, the observed retention of BTH-induced NPR1 nuclear accumulation (Fig. 4), and the requirement for NPR1 and the TGA2, TGA5, TGA6 TFs (Fig. 7e, f) indicates that the core SA signalling pathway is utilized to facilitate this resistance at 30 °C. Global transcriptome analysis of BTH-regulated genes revealed a temperature-sensitive bifurcation in the SA signalling pathway at the 24 h time point assessed in our study, with ~60% of genes in the canonical *PR1/ICS1* temperature-sensitive branch, and ~40% of genes in the temperature-insensitive branch (Figs. 5 and 8). Based on the lack of induction of *PR1* and *ICS1* by either *Pst* DC3000 or BTH alone (Figs. 2a and 3a) or in combination at 30 °C (Fig. 7c), it is possible that a major part of the SA-regulated transcriptome is dispensable for BTH-mediated protection against *Pst* DC3000 (Fig. 8). This surprising result indicates that either the temperature-insensitive branch of the SA transcriptome and/or a gene expression-independent process underlies BTH-mediated

resistance against *Pst* DC3000 at 30 °C. We chose the 24 h time point for gene expression analysis because this is when *PR1*, *PR2* and *PR5* genes are most highly expressed following BTH treatment[36]. However, it is important to note that this provides only a snap shot of the dynamic transcriptional landscape, and future research should examine whether temperature-sensitive genes may be induced or suppressed at earlier or later time points.

In summary, we have studied the impact of an important climate condition, elevated temperature, on a widely studied compatible plant–pathogen interaction. Our results highlight pathogen-induced SA production as a key temperature-vulnerable step in the SA defence network. In addition, our study revealed a surprisingly positive effect of elevated temperature on bacterial translocation in planta, which challenges the long-standing notion based on previous in vitro studies that an efficient T3SS requires a low temperature (e.g., 18–20 °C)[8]. Surprisingly, in spite of the increase in pathogen virulence, we still observed BTH-mediated protection against *Pst* DC3000 infection. This finding supports the continued use of BTH as a crop protectant even at elevated temperatures. Because the SA pathway is an integral component of the plant immune system, we hope that the fundamental insights gained from this study will stimulate future research to uncover additional temperature-sensitive and temperature-insensitive nodes of the plant immune system. This information should prove useful for genetic manipulation of climate-relevant components of the plant immune system to enhance plant resilience to combined adverse abiotic and biotic conditions.

## Methods

**Plant materials and growth conditions**. *Arabidopsis* Columbia-0 (Col-0), Landsberg *erecta* (L*er*) WT and mutant plants (in Col-0 or L*er* background) were soil-grown (2:1 'Arabidopsis mix':perlite covered with standard Phiferglass mesh) for 3–4 weeks at 12 h light ($85 \pm 10 \, \mu mol \, m^{-2} \, s^{-1}$), 12 h dark, 23 °C and 60% relative humidity. The *pif1-1 pif3-3 pif4-2 pif5-3* (*pifq*)[29], *tga2-1 tga5-1 tga6-1*[59], *sid2-2* (*ics1*)[11], *ein2-1*[55], *ein3-1 eil1-1*[48], *dde2-2* (*aos*)[53], *myc2 myc3 myc4*[54], *aba2-1*[52] and *camta2 camta3-1*[56] mutant plants and 35S::*PHYB*[Y276H] (YHB)[31,32] transgenic plants were previously characterized. The *npr1-6* (SAIL-708F09) T-DNA insertion mutant was obtained from the *Arabidopsis* Biological Resource Centre (ABRC) at The Ohio State University. T-DNA insertion mutants were genotyped using the REDExtract-N-Amp Plant PCR kit (Sigma) following the manufacturer's protocol. Primers used for genotyping are listed in Supplementary Table 6.

**Temperature and chemical treatments**. For temperature assays, chambers were set to either 23 °C (control) or 30 °C (test), with all other conditions the same as above. Three-week-old to four-week-old plants were used for all experiments except confocal microscopy, for which 2-week-old plants were used to minimize loss of age-related decrease in YFP-associated fluorescence. Plants were moved to test chambers 2 h after lights were turned on and acclimated for 48 h before pathogen infiltration. For experiments with chemical pre-treatment, plants were temperature-acclimated for 24 h before spraying with either mock (0.1% DMSO, 0.01% Silwet L-77) or benzo(1,2,3)thiadiazole-7-carbothioic acid-S-methyl ester (BTH, Chem Service Inc.; 100 μM, 0.1% DMSO, 0.01% Silwet). For callose assays, flg22 (200 nM in 0.1% DMSO) served as a positive control, and all solutions were infiltrated into leaves using a needleless syringe. Pathogen infection or other assays were performed 24 h after chemical treatment.

**Phytohormone extraction and quantification**. Phytohormones were extracted and quantified as previously described[66] with some modifications. Leaf tissue between 10 and 50 mg (fresh weight, FW) was flash-frozen in liquid nitrogen, ground and extracted at 4 °C overnight (~16 h) using 0.3–0.5 mL of ice-cold extraction buffer (methanol:water (80:20 v/v), 0.1% formic acid, 0.1 g L$^{-1}$ butylated hydroxytoluene, 100 nM ABA-d$_6$). Filtered extracts were quantified using an Acquity Ultra Performance Liquid Chromatography (UPLC) system (Waters Corporation, Milford, MA) as previously described[66], except the capillary voltage, cone voltage and extractor voltage were set to 3.5 kV, 25 V and 5 V, respectively, and the desolvation gas and cone gas were set to flow rates of 600 L h$^{-1}$ and 50 L h$^{-1}$, respectively. Selected ion monitoring (SIM) was conducted in the negative ES channel for SA ($m/z$ 137 > 93), SA glucoside (SAG; $m/z$ 299.1 > 137), ABA ($m/z$ 263.1 > 153.1) and the internal ABA-d$_6$ standard ($m/z$ 269.1 > 159.1). Parent > daughter SIM pairs, as well as the optimal source cone and collision energy voltages for each compound monitored were determined using Quan-Optimize software. Analyte responses based on peak area integrations relative to the internal standard was determined using QuanLynx v4.1 software (Waters,

Milford, MA). Both the SA and SAG analytes were quantified based on the SA standard curve and ABA was quantified based on the ABA standard curve to calculate the sample concentrations (nM), which were converted to ng using the molecular weight of the compound and the extraction volume, and were then normalized by sample FW in g. SA and SAG concentrations were combined and reported as total SA.

**RNA extraction and qPCR**. RNA was extracted from flash-frozen, ground leaf tissue with the ToTally RNA kit following the manufacturer's protocol (Ambion). Samples were digested with DNaseI (Roche) to remove any genomic DNA contamination, and then purified using the RNeasy Mini kit (Qiagen). M-MLV reverse transcriptase (RT, Life Technologies) was used to synthesize cDNA. For all genes of interest, ~1.5 ng of cDNA template was used for quantitative PCR (qPCR), with expression normalized to the *PP2AA3* internal control gene using the equation $2^{-\Delta CT}$, where $\Delta C_T$ is $C_{T \, target \, gene} - C_{T \, PP2AA3}$ (see Supplementary Table 6 for primer sequences). All qPCR reactions were performed using the SYBR® Green master mix (Life Technologies) and 7500 Fast Real-Time PCR system (Applied Biosystems, Foster City, California), with three technical replicates and a minimum of three biological replicates per experimental treatment.

**RNA sequencing and data analysis**. Three biological replicates of each experimental treatment were selected based on quality (RIN score 7) using a BioAnalyzer Agilent 2100. Samples were pooled on two lanes of an Illumina HiSeq 2500 Rapid Run flow cell (v1) and sequenced in a 1 × 50 bp single-end format using Rapid SBS reagents. Details regarding sequencing, read counts and data analysis are provided in Supplementary Note 1.

**Callose accumulation**. Following temperature acclimation and chemical treatment, leaves were harvested and cleared in 100% ethanol overnight. Cleared leaves were fixed with a 75% ethanol, 25% acetic acid solution for 2 h, after which leaves were washed consecutively with 75% ethanol, 50% ethanol and 150 mM K$_2$HPO$_4$ pH 9.5 for 15 min. Finally, leaves were stained in an aniline blue solution (0.1%, 150 mM K$_2$HPO$_4$ pH 9.5) overnight at 4 °C. Callose deposits were visualized using an Olympus IX71 inverted microscope with a 120-watt metal halide lamp (X-Cite series 120) and a DAPI filter (Semrock, excitation 377/50 and emission 447/60). Images shown are at ×10 magnification. Callose counts were processed using ImageJ (Rasband W.S., National Institutes of Health, USA). Images were first converted to 32-bit grey-scale, after which the threshold of the image was adjusted so that only callose deposits were visible over the background. Callose deposits were then counted using the analyse particles tool. Four callose measurements were collected per leaf; each individual leaf was collected from a different plant. Six to eight plants were evaluated per treatment.

**Preparation and selection of transgenic lines**. The pNPR1::NPR1-YFP and p35S::YFP constructs were generated by Gateway cloning and then used for *Agrobacterium*-mediated transformation of *npr1-6* (SAIL_708F09) and Col-0 plants. Details for cloning and screening of transgenic lines are described in Supplementary Note 2.

**Disease and BTH protection assays**. *Pst* DC3000 inoculum was prepared at a concentration of ~1–3 × 10$^6$ colony-forming units (CFU) mL$^{-1}$ (an optical density at an absorbance wavelength of 600 nm (OD$_{600}$) of ~0.001) as described in Supplementary Note 3. For time course disease assays with *ics1* and *pifq* mutants, a lower inoculum of ~4–5 × 10$^5$ was used to assess disease phenotypes in the mutant plants prior to saturation of bacterial growth. Bacterial infiltration and quantification are described in Supplementary Note 3.

**In vitro growth assay**. An overnight culture of *Pst* DC3000 (grown in LM + Rif) was diluted to an OD$_{600}$ of 0.05 (LM only), which was then divided into six 125 mL flasks. Three flasks each were incubated with shaking (225 RPM) at 23 °C or 30 °C. Bacterial populations were assessed at each time point by measuring the OD$_{600}$ with a spectrophotometer (DU 800, Beckman Coulter).

**Effector translocation assay**. Following temperature treatments, plants were infiltrated using a needle syringe with a high inoculum (2–4 × 10$^7$ CFU mL$^{-1}$, 0.25 mM MgCl$_2$) of *Pst* DC3000 or the *hrcC*$^-$ mutant carrying the P$_{nptII}$::*avrPto-CyaA* plasmid[35], or *Pst* DC3000 containing either the P$_{tac}$::*avrPtoB-CyaA*[67], P$_{tac}$::*hopU1-CyaA*[67] or P$_{tac}$::*hopG1-CyaA*[68] plasmid. For translocation experiments using plants pre-treated with mock or BTH, BTH-treated plants were infiltrated with inoculum ~1.5 times higher than that used to infiltrate mock-treated plants. This was done to enable samples with equivalent bacterial populations 4 hpi. Leaf discs were harvested using a biopsy punch 4–6 hpi for both bacterial population quantification and cAMP quantification, which was normalized by total plant protein. cAMP was extracted and quantified using the Direct cAMP ELISA kit (ENZO) according to the manufacturer's protocol. Total protein was quantified using a Quickstart Bradford assay (BioRad) according to the manufacturer's protocol.

**Nuclear fractionation and western blotting**. Following temperature and chemical treatments, a minimum of 0.5 g (FW) leaf tissue was harvested and the mass recorded prior to flash freezing in liquid nitrogen. After grinding, cell lysate was isolated using the CelLytic PN Isolation/Extraction Kit (Sigma), using the semi-pure fractionation method according to the manufacturer's protocol. Following isolation and fractionation, the whole-cell lysate and cytosolic fractions were diluted with an equal volume of 4× SDS Laemmli sample buffer (125 mM Tris-HCl, pH 6.8; 4% (w/v) SDS; 20% glycerol; 0.02% bromophenol blue; 5% (v/v) β-mercaptoethanol) while the nuclei pellet was resuspended in 100 μl of 2× SDS Laemmli sample buffer, resulting in a nuclear-fraction sample with an 8-fold higher concentration than the whole-cell or cytosolic-fraction samples. All samples were boiled for 10 min at 95 °C, equal volumes of each were loaded in 4–12% SDS-PAGE gradient gels (NuPAGE, Novex) and run for 40–50 min at 200 V. Proteins were transferred to PVDF membranes at room temperature for 1 h at 25 V. Primary antibodies used were α-GFP (1:7500, Abcam, cat# ab290), for detection of NPR1-YFP; α-PR1[69] (1:5000, gift from Xinnian Dong, Duke University); α-UGPase (1:3000, Agrisera, cat# AS05 086) and α-H3 (1:10,000, Agrisera, AS10 710). The secondary antibody used for all blots was a goat α-rabbit (1:20,000, ThermoFisher Scientific, cat# 31460). A third protein gel was run and stained with Coomassie to enable visualization of protein loading. Uncropped images for all western blots are provided in Supplementary Fig. 16.

**Confocal microscopy**. Images were taken on a Zeiss 510 Meta Confocal Laser Scanning system configured on a Zeiss AxioObserver.Z1 inverted microscope (Carl Zeiss Microscopy, Thornwood, NY) using a Zeiss ×63 C-Apochromat water immersion objective (NA 1.2). Sequential imaging with a Kalman averaging of 4 and pinhole set to 120 μm was used to capture images from a single confocal plane. Bright field (BF) images were recorded using an Argon 514 nm laser. YFP was excited with an Argon 514 nm laser set at 20% and fluorescence emission was recorded using a 535–565 band pass filter. Aim Image Browser Software (Zeiss LSM) was used to add scale bars and for adjustments to brightness and contrast, which were as follows: NPR1-Y1 images, contrast 60%; npr1 images, contrast 65%; 35S::YFP YFP images, brightness 52%, contrast 55%; 35S::YFP BF images, brightness 48%, contrast 55%. Images were also adjusted for sharpness and contrast using Microsoft PowerPoint as follows: all images, sharpness + 50%; npr1 BF images, contrast (+20%); 35S::YFP YFP images, −40% contrast; 35S::YFP BF images, +20% contrast.

**Statistical analysis**. A minimum of three independent experiments were done for all assays unless otherwise indicated. Sample size used for experiments is indicated in the figure legends and was selected based on previously published work of similar experiments shown to be sufficient for statistical analyses. Statistical significance was determined using a Student's $t$ test (Excel) for pairwise comparisons, a one-way analysis of variance (ANOVA) followed by a Dunnett's test (Prism 6, GraphPad Software, Inc.) for comparisons of multiple test samples to the same control, or by conducting a 2 × 2 factorial ANOVA followed by Tukey's honest significant difference (HSD) test (RStudio (https://www.rstudio.com/)) for multivariate analyses. In the case of unequal variances as determined by the Brown–Forsythe test ($\alpha \leq 0.05$, Prism 6, GraphPad Software, Inc.), data were log-transformed prior to conducting statistical analysis.

**Data availability**. All the RNA-seq reads for this work have been submitted to NCBI under BioProject PRJNA325245. The authors declare that all other data supporting the findings of this study are within the manuscript and its supplementary files or are available from the corresponding author upon request.

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

## Acknowledgements

We thank Dr. A. Daniel Jones for the (ABA)-d$_6$ standard used for LCMS; Dr. Xinnian Dong for the α-PR1 antibody; Dr. Peter Quail for the *pifq* mutant; Dr. Gregg Howe for the *myc2 myc3 myc4* mutant; Dr. Michael Thomashow for the *camta2 camta3* mutant; Dr. Clark Lagarias for the *YHB* transgenic line; Dr. Brian H. Kvitko for generation of the *Pst* DC3000(P$_{nptII}$::*avrPto-CyaA*) and *Pst* DC3000 *hrcC⁻*(P$_{nptII}$::*avrPto-CyaA*) strains; Dr. James Alfano for the *Pst* DC3000(P$_{tac}$::*avrPtoB-CyaA*), *Pst* DC3000(P$_{tac}$::*hopU1-CyaA*) and *Pst* DC3000(P$_{tac}$::*hopG1-CyaA*) strains; Melinda Frame for confocal microscopy support; Jim Klug and Cody Keilen for greenhouse support; and Scott A. Caughel for artistic rendering of the model used in Fig. 8. We also acknowledge funding provided by the College of Natural Science, Michigan State University Dissertation Continuation Fellowship, Plant Science Excellence Fellowship and University Enrichment Fellowship to B.H.; funding from the Max Planck Society and Deutsche Forschungsgemeinschaft grant SFB670 to K.T.; support from the Chemical Sciences, Geosciences and Biosciences Division, Office of Basic Energy Sciences, Office of Science, US Department of Energy through grant no.DE–FG02-91ER20021 to B.L.M. and support from the Gordon and Betty Moore Foundation GBMF3037 and Michigan State University Plant Resilience Institute to S.Y.H.

## Author contributions

B.H., B.L.M. and S.Y.H. conceived research and designed experiments. C.D.M.C. contributed to qPCR, disease assays and translocation assays. A.V.L. performed callose experiments. E.H. contributed to qPCR and complementation analysis of *pNPR1::NPR1-YFP* transgenic lines. For RNA-seq analyses, J.A.P. conducted DAVID analyses, K.L.C. assisted with gene expression profiling and K.T. performed DEG, k-means clustering, heat map generation, Genevestigator and AME analyses. J.Y. generated the *p35S::YFP* transgenic line, performed initial characterization of the *npr1-6* allele and assisted with generation of the *pNPR1::NPR1-YFP* transgenic lines. B.H. performed all other experiments. B.H. and S.Y.H. wrote the manuscript with input from all co-authors.

## Additional information

**Competing interests:** The authors declare no competing financial interests.

