## [Peer Review File · Nature Communications]

Reviewers' comments:

Reviewer #1 (Remarks to the Author):

This manuscript by Huot et al. describes the effect of higher temperature (30 °C) on SA-mediated defense and disease resistance against *Pseudomonas syringae*, among other interesting analyses in an attempt to identify key players in the temperature modulation of defense. Compared to 23 °C, growth acclimation at 30 °C promotes Pst DC3000 infection and this is independent from PIFs. By using Pst T3SS and COR mutants, authors identify that enhanced growth of Pst at 30 °C depends on the presence of the T3SS but not COR. Furthermore, higher avrPto effector delivery is detected at 30 °C compared to 23 °C. Authors find that loss of SA pathway is compromised at higher temperature, and the *ics1* mutant is insensitive to temperature. Higher disease susceptibility to Pst DC3000 infection at 30 °C is due both to promotion of effector translocation and loss of SA biosynthesis, but independent from ET, JA and ABA, which are known to antagonize SA biosynthesis and signaling.

Authors then use BTH to establish a pathogen-free system for further analyses. SA-marker PR1 and ICS1 induced expression by BTH as well as callose deposition are impaired at 30 °C, but not flg22-triggered callose deposition.

Authors then analyze the subcellular localization of NPR1 protein, a known regulator of SA signaling, by confocal microscopy and subcellular fractionation. BTH treatment increases NPR1 protein levels, but temperature or BTH treatment do not affect NPR1 localization. It is suggested that the temperature effect should be downstream NPR1. Through RNAseq analyses, authors identify sectors sensitive to temperature and/or BTH. Motif enrichment analyses pointed to CAMTA transcription factors, which were studied using the *camta2/3* double mutant. This mutant exhibits higher resistance to Pst DC 3000 at 23 °C and 30 °C, compared to the wild-type. ICS1 and PR1 expression are compromised, but SA levels are higher at either temperature. Authors conclude that CAMTA are not responsible for loss of SA induction at higher temperature. Last, the protective effect of BTH against Pst DC3000 infection at high temperature is shown, despite the lack of induction of the PR1/ICS1 branch determined by RNAseq. However, this is dependent on the presence of a functional NPR1 and TGA2/5/6 (core components of the SA pathway).

Comments:

The amount of work reported in this manuscript is important and I appreciated this. However, I missed the identification of specific genes or clearly defined mechanisms responsible for the temperature modulation of defense in this pathosystem. The results discard the involvement of PIF transcription factors, ethylene, jasmonates and ABA biosynthesis or signaling, and CAMTA 2/3. Results are conclusive for the need of known core components of the SA signaling NPR1 and TGA2, TGA5 and TGA6.

Introduction

1. The link with climate change of this manuscript could be improved. Here, authors report the effect of constant temperatures (23 °C vs 30 °C) on bacterial virulence. In real scenarios, the temperature fluctuates during the day and night. The two references in line 46 supporting this link are from years 1928 and 1973. In none of them there's a reference to climate change but environmental effects on disease, mainly from an agricultural perspective.

2. Line 75 states that over-expression of NPR1 improves fitness in field-grown *Arabidopsis* (ref. 19). However, in this work seed yield could not be tested in the field, and rosette diameter was used as proxy for fitness. In addition, authors reported that no difference in fitness between overexpressor NPR1 and wild-type could be found.

Results

3. In Mat & met authors describe that plants were transferred back to test chambers (at 23 °C or 30 °C) after inoculation with different *Pseudomonas syringae* strains. It seems to me a bit unclear how to distinguish the temperature effects on bacterial growth from disease susceptibility after 3 days at contrasted temperatures. Supp Fig. 2 b shows clear differences on bacterial growth by temperature after 24 h.

4. What's the effect on disease resistance of 30 °C pre-acclimated plants which are transferred back to 23 °C after inoculation? Is the temperature pre-acclimation sufficient for higher susceptibility?

5. Line 162 (and elsewhere in the manuscript) says that "the *ics1* mutant was not temperature sensitive" but this sentence is not accurate. It refers to the temperature effect on SA-pathway. Does the *ics1* mutant show the morphological responses to high temperature?. The *ics1* mutant is not insensitive to temperature, as a general statement. It is impaired in SA biosynthesis, which mimics the high temperature effect on SA pathway. This general statement is confusing.

6. What "divergence" refers to in line 145?

7. Authors test the effect of higher temperature on the translocation of one effector by monitoring *avrPto-cyaA* delivery. They observe increased translocation at 30 °C compared to 23 °C. Authors conclude that "30 °C is linked to increased translocation of bacteria effector proteins". However, this cannot be generalized to all effectors, unless a significantly higher number of effectors is tested.

8. Results in Fig. 2e-h would be more informative if they are complemented with Pst DC3118 and *hrcC* strains in the expression analyses of ET, JA and ABA biosynthesis/signaling genes.

9. Protein load should be quantified in Figure 4b. Currently it only refers to loading of "equal volumes" but the high amounts of protein detected by the anti-H3 antibody do not enable to clearly assess equal loading between samples. In addition, the approx size of bands detected in the western blots should be indicated in the figures.

Reviewer #2 (Remarks to the Author):

The manuscript addresses potential mechanisms that are responsible for the higher susceptibility of *Arabidopsis* plants towards the virulent *Pseudomonas* strain DC3000 when grown at 30°C. First, the authors showed that enhanced translocation of the effectors cannot be the major reason for the increased susceptibility. After showing that the synthesis of SA is compromised at 30°C, the authors excluded candidate PIFs as negative regulators of immunity at elevated temperatures. The dominant question, which heat labile component (PhyB?) regulates the activation of SA biosynthesis was not further addressed. Instead it was analyzed, whether the chemical resistance induced by the SA-analogue BTH is compromised as well. Interestingly, BTH-induced resistance was not compromised at 30°C. Microarray analysis of BTH-treated plants at 23°C and 30°C identified a set of NPR1-regulated genes that were sensitive to 30°C, and another set that was not sensitive. Overall, the manuscript provides a wealth of novel data that are interesting for the community.

I have only a few comments that need to be corrected:

1. The roles of GRXS13 and GRX480 as positive regulators of SA signaling are not yet documented, references 30 and 31 do not prove this. These genes are just induced by SA. The role of GRX480 as an SA-induced negative regulator of JA/ET signaling has not yet been confirmed by loss of function analysis.

2. Supplemental Table listing the 2844 DEGs: it is not easy for the reviewer to evaluate the quality of the RNAseq data: At least, the p-values should be indicated. Page 14: For the analysis based on differences of expression levels, which p value was chosen for the cut-off? It would be appreciated to have a list of group A genes, group B genes, etc along with the expression values.

Reviewer #3 (Remarks to the Author):

The manuscript "Dual impact of elevated temperature on plant defence and bacterial virulence in *Arabidopsis*" by Hout et al. focuses on plant immune responses in conjunction with increased temperature. Under such conditions, the authors demonstrated heightened susceptibility of *Arabidopsis* plants that is independent of PIFs and dependent upon pathogen virulence factors (effectors). Detailed transcriptome analyses revealed that SA-mediated signaling components including ICS1 are impeded. Exogenous application of SA analog can rescue these phenotypes. This manuscript builds upon two previous publications: The first article showed that increased temperature modulates basal and effector-triggered immunity (Wang et al 2009 MPMI). Moreover, this article suggests the involvement of SA as well as two major players of basal defense and ETI, EDS1 and PAD4, in temperature-dependent increase in disease susceptibility. Cheng et al (Nature Comm (2013) in the second article performed MTI and ETI assays with increasing intensities of temperature and demonstrated that elevated temperature promotes and inhibits MTI and ETI, respectively. In the light of the existing information, the current manuscript focuses on understanding the underlying molecular mechanisms that lead to the increased host susceptibility by a virulent pathogen. Major points that need to be taken into account are as follows:

1- The authors demonstrated that DC3000 and DC3118 but not HrcC caused increased susceptibility at the elevated temperature that is independent of PIFs. First, the authors talked about climate change and its impact on plant immunity and agricultural yield. Yes, climate change is real and it's happening but one cannot realistically expect a dramatic 7C change in temperature that will favor bacterial growth, while plants never got the opportunity to co-evolve. I understand one can't fast forward in time and predict how the plant genome would shape up by the time there will be a 7C increase in temperature. However, one could perform experiments with an increasing magnitude of temperature to understand plant genome plasticity and figure out the minimum increase in temperature that would shift the balance and allow for compromised basal defense. I'm also puzzled why the authors moved the plants for 48 hrs prior to pathogen infection unlike Cheng et al where they moved the plants into increased temperature conditions 15 minutes prior to the experiment? I'm wondering about a quick water loss since the plants showed hyponastic symptoms. How about the "heat shock" response? The authors have performed global transcriptome but I did not see data corresponding to 23C and 30C transcriptome changes? What is the status of stomatal aperture? Should the authors consider using spray or dip inoculation as opposed to syringe infiltration to assay if stomatal immunity is intact? The authors claimed that "PIFs may not play a major role in mediating elevated temperature-dependent enhancement of disease susceptibility to Pst DC3000". However, the growth of bacteria on day 3 reached 10^8 . Could the authors use a relatively low dose of bacteria to start with? Alternatively, they can monitor the dynamic growth of pathogens by taking samples every 24 hrs. The authors used PnptII::avrPto-cyaA reporter construct to assess the translocation rate at a higher temperature. They found a three-fold increase in the translocation activity. However, they generalized the conclusion that "increased Pst DC3000 virulence at 30°C is linked to increased translocation of bacterial effector proteins". Is this the only effector that is translocated rapidly or most of the effectors can be translocated faster? Are they more stable and/or accumulate more protein at a higher temperature? Given that MTI is induced at a higher temperature, it would be important to test delivery of ETI-triggering effectors such as avrB, avrRpm1, avrRpt2, avrPphB into the mutants that can't trigger corresponding NLR-mediated resistance at a higher temperature to investigate roles of these effectors in the MTI suppression.

2- The authors checked the expression patterns of ICS1 and PR1 upon infection with DC3000 (vacuum infiltration) and showed that the levels of expression and total cellular SA concentration are diminished at elevated temperatures. I would be intrigued to see such expression study performed at different temperature conditions as well as different time points since SA-dependent genes and SA accumulation are under circadian control. What if the authors use spray inoculation, ETI inducing DC3000 strain or HrcC? Similar to my previous comment about infection experiment on the *pif* mutants, the authors should consider using low dose and/or early time points to monitor the growth of bacteria in *ics1* mutant.

3- Diminished accumulation of SA at increased temperature is the most direct evidence in this manuscript. Given that SA levels are reduced and MYC genes are induced, it would be valuable to examine the growth of necrotrophic pathogens at higher temperature.

4- The authors focused on the master regulator of SA pathway, NPR1 and showed that NPR1's translocation to the nucleus is not affected. Given that SA is required for NPR1's nuclear translocation and SA is not accumulated at 30C, the question would be how NPR1 is translocated to the nucleus? Does heat alter the redox status of the cell?

5- The authors performed RNA-seq experiment and identified over 2,800 genes that are BTH-regulated and 30C-dependent. They analyzed 23°C BTH vs. 30°C BTH) and (ii) fold change (BTH/mock). I wonder if the PI also analyzed 23 C and 30 C mock data to dissect the landscape of transcriptional changes that occur due to the change of temperature. The authors made four groups including BTH-induced but 30C low expression, BTH-induced 30C up expression and BTH-suppressed and 30C less suppressed. It seems that the amplitudes of 66% BTH-regulated genes is altered. The authors have performed all the qRT-PCRs and RNA-seq using only one time point (24 hours). Based on these experiments, the authors suggest an SA-mediated bifurcation or qualitative topological behavior of these genes. The authors did not specify an analysis that led them to make this conclusion. What if the amplitude of the same set of genes may vary at different time points within a classified group? Keeping in mind most of these genes are under circadian control, it would be interesting to examine the transcript levels of a set of genes, at least the one in figure 5, with a fine time course qRT-PCR. The authors acclimated the plants for 24 hours prior to BTH spray. I was wondering whether the authors would obtain the same results if they moved the plants right after BTH spray to 23C and 30 C for qRT-PCR similar to what Cheng et al have done. The authors performed K cluster analysis to group set of genes with similar expression at 24 h. Subsequently, they performed GO ontology and element enrichment analyses. It was puzzling to me how genes in clusters 2, 4 and 5 can be regulated by TGA factors simultaneously (based on the assumption that TGAs are BTH-induced but 30C low expression; group A blue genes) since they are categorized in group A, B and C. In theory, it is possible if TGAs are interacting with a set of positive or negative regulators that may positively and negatively influence the expression patterns. As it stands, however, the authors did not provide any genetic or biochemical evidence (totally hand waving). For instance, performing q-RT PCR on genes from cluster 2, 4 and 5 in *tga2/5/6* mutant background. The same argument is true for WRKY and PIFs given that PIFs are not even involved in temperature-mediated disease susceptibility. Similarly, the authors just checked ICS1 and PR1 in *camta* mutant. How about testing the genes from cluster 6?

The authors refuted model 1 by stating that they observed increased translocation of *avrPto* at 30C and *hrcC* growth at 30C is not increased. Given that SA accumulation is dampened at 30C, would the authors perform effector translocation assay on SA-sprayed plants at 30C with the same results? In model 3, the authors suggested that other hormones or hormonal nodes may suppress SA biosynthesis. I was wondering why the authors did not perform SA measurements or measured transcript accumulation of ICS1, their hallmark marker gene, in *aos*, *myc2/3/4*, *ein2* etc. mutants.

Responses to Reviewer #1:

Thank you for taking the time to thoughtfully review our manuscript. Please see our responses to your comments below.

General Comments:

The amount of work reported in this manuscript is important and I appreciated this. However, I missed the identification of specific genes or clearly defined mechanisms responsible for the temperature modulation of defense in this pathosystem. The results discard the involvement of PIF transcription factors, ethylene, jasmonates and ABA biosynthesis or signaling, and CAMTA 2/3. Results are conclusive for the need of known core components of the SA signaling NPR1 and TGA2, TGA5 and TGA6.

Response: You are correct. We have not identified a specific gene that “switches” temperature effects on plant defense signaling and pathogen virulence. Identifying such a specific gene/switch would be truly wonderful. However, our results so far show that elevated temperature actually has multiple effects on plant disease development (e.g., on multiple plant defense signaling pathways and on pathogen virulence systems), and may or may not involve a simple switch. In particular, we identified the positive regulation of the type III secretion system (T3SS; a central bacterial virulence system) as a major mechanism that underlies the temperature effect on increased disease susceptibility (see p 8 – 9, lines 160 – 172). This surprising finding challenges the widely held view, based on previous *in vitro* studies, of the negative effects of elevated temperature on the T3SS. Please note that the perceived negative effects of elevated temperature on the T3SS were used in a recent *Nature Commun* paper to formulate the most recent model on temperature effects on bacterial infection¹. If our results had been available, we believe that Dr. Ping He and Dr. Libo Shan would likely have proposed a conceptually different model. In this regard, our study provides a very important new insight.

Furthermore, as you correctly pointed out, we observed that elevated temperature impacts multiple plant defense signaling pathways, and conducted experiments to show that PIF transcription factors, ethylene, jasmonates and ABA biosynthesis or signaling, and CAMTA 2/3 are not involved (see p 7, lines 134 – 140; p 15, lines 330 – 340, p 16, lines 352 - 356). Instead, we show that reduced salicylic acid production is a major mechanism that underlies the temperature effect on increased disease susceptibility (see p 9, lines 181 – 191). We did not stop here, but went on to identify, for the first time, temperature-sensitive and temperature-insensitive branches of the SA gene expression network (see p 12 – 14, lines 266 – 291). We found that the commercial SA-analogue, BTH, can confer robust protection against disease at elevated temperature in the absence of canonical SA-responsive *ICS1* and *PR1* gene expression (see p 16, lines 363 – 368), a finding that should have significant practical implications in disease control.

We think that these results offer substantial new insights into how elevated temperature could impact a compatible/susceptible plant-pathogen interaction, which involves multiple players (as opposed to previous studies on single *R*-gene-mediated resistance or single PAMP-triggered plant immunity). We have modified text in the revised manuscript to highlight these statements (page 17-18, 1st paragraph of Discussion). In addition, we followed your suggestion and conducted further experiments to strengthen this manuscript (see below).

Introduction Comments:

1. *The link with climate change of this manuscript could be improved. Here, authors report the effect of constant temperatures (23 °C vs 30 °C) on bacterial virulence. In real scenarios, the temperature fluctuates during the day and night. The two references in line 46 supporting this link are from years 1928 and 1973. In none of them there's a reference to climate change but environmental effects on disease, mainly from an agricultural perspective.*

Response: This is another good point. Clearly, our research was not designed to directly address the climate change issue. Instead, we wanted to investigate the effect of elevated temperature on one of the best studied compatible (susceptible) plant-pathogen interactions, on which thousands of papers have been published, but with little insight into how a single environmental variable, such as temperature, would affect the disease outcome. We have replaced the 1928 paper with a more recent reference to highlight the increasing concerns of climate change (see p 4, line 47), but have otherwise removed unnecessary references to climate change. In light of the profound effect of even a simple temperature variation on a compatible plant-pathogen interaction, as shown in this study, we hope that our study may inspire future research by many groups to explore a variety of temperature regimes, including fluctuating temperature conditions, on disease susceptibility.

2. *Line 75 states that over-expression of NPR1 improves fitness in field-grown Arabidopsis (ref. 19). However, in this work seed yield could not be tested in the field, and rosette diameter was used as proxy for fitness. In addition, authors reported that no difference in fitness between overexpressor NPR1 and wild-type could be found.*

Response: We apologize for this incorrect statement and have removed this statement along with the associated reference.

Results Comments:

3. *In Mat & met authors describe that plants were transferred back to test chambers (at 23 °C or 30 °C) after inoculation with different Pseudomonas syringae strains. It seems to me a bit unclear how to distinguish the temperature effects on bacterial growth from disease susceptibility after 3 days at contrasted temperatures. Supp Fig. 2 b shows clear differences on bacterial growth by temperature after 24 h.*

Response: We actually did experiments to address the effect of temperature on bacterial growth *in vitro* and saw no difference at 23 °C vs. 30 °C. We should have included this data in the original submission to make it clear that the temperature effects were observed only when bacteria are inside the plant. In response to your comment, we have now added this data to the revised manuscript (new Supp Fig 2; page 7, lines 120 – 122).

The previous Supp Fig. 2b was to show that bacterial growth differed at 23 °C vs. 30 °C, as expected. However, we realize that this is not relevant for the effector translocation assays, which were performed at much earlier time points (4 to 6 hpi) when bacterial populations have not diverged. Therefore, we have removed the 24 hpi data from this particular Supplementary figure. Thank you.

4. What's the effect on disease resistance of 30 °C pre-acclimated plants which are transferred back to 23 °C after inoculation? Is the temperature pre-acclimation sufficient for higher susceptibility?

Response: We conducted new experiments to comprehensively address this issue. Specifically, we conducted disease assays in plants that were (i) acclimated and kept at 23 °C following infection (23 °C → 23 °C), (ii) acclimated at 30 °C and shifted to 23 °C (30 °C → 23 °C), (iii) acclimated at 23 °C and shifted to 30 °C following infection (23 °C → 30 °C), or (iv) acclimated and kept at 30 °C following infection (30 °C → 30 °C). We observed that the post-infection temperature is more important than the pre-acclimation temperature. However, the magnitude of the difference in bacterial growth was more consistently greater in plants pre-acclimated to elevated temperature. These results deepen our understanding of the temperature effect and are added to the revised manuscript (new Supplementary Fig 3; page 7, lines 123 – 131).

5. Line 162 (and elsewhere in the manuscript) says that “the ics1 mutant was not temperature sensitive” but this sentence is not accurate. It refers to the temperature effect on SA-pathway. Does the ics1 mutant show the morphological responses to high temperature? The ics1 mutant is not insensitive to temperature, as a general statement. It is impaired in SA biosynthesis, which mimics the high temperature effect on SA pathway. This general statement is confusing.

Response: Thank you for pointing this out. Yes, the *ics1* mutant exhibits the typical morphological response to temperature. We have changed the language in the manuscript to more specifically state that the *ics1* mutant exhibits a temperature-insensitive disease phenotype (see page 18, lines 405 – 406).

"... the SA-deficient *ics1* mutant exhibits a largely temperature-insensitive disease phenotype, with a similar level of susceptibility as WT plants at 30 °C (Fig. 2c)."

6. What “divergence” refers to in line 145?

Response: “Divergence” here refers to when bacterial populations start to differ at different temperatures. We have modified the text accordingly to improve clarity (see p 8, lines 164-167).

“To ensure differences in effector translocation were not influenced by bacterial populations, plant samples were first selected based on having similar bacterial populations at the 4 – 6 hours post infiltration (hpi) time point used for the translocation assay (Supplementary Fig. 5a, b).”

7. Authors test the effect of higher temperature on the translocation of one effector by monitoring *avrPto-CyaA* delivery. They observe increased translocation at 30 °C compared to 23 °C. Authors conclude that “30 °C is linked to increased translocation of bacteria effector proteins”. However, this cannot be generalized to all effectors, unless a significantly higher number of effectors is tested.

Response: We have followed your suggestion and conducted effector translocation assays with three additional effector-CyaA strains, HopU1-CyaA, HopG1-CyaA and AvrPtoB-CyaA, available from published studies. Translocation of all these effectors showed a similar trend as was observed with AvrPto-CyaA

(See revised Fig 1e, page 8-9, lines 162 - 172). This result strengthens the conclusion.

8. *Results in Fig. 2e-h would be more informative if they are complemented with Pst DC3118 and hrcC strains in the expression analyses of ET, JA and ABA biosynthesis/signaling genes.*

Response: Assessing ET, JA and ABA marker gene expression in plants infiltrated with either *Pst* DC3118 or *hrcC* mutant strains would provide further information about whether the observed induction was dependent on ET, JA and ABA biosynthesis/signaling genes. However, since we already observed that ET, JA or ABA hormone mutants exhibit enhanced susceptibility to disease at elevated temperature similar to wild type plants (see new Fig. 6b, d, f and Supplementary Fig 14b, d), we feel that further experiments along this line would likely not lead to biologically relevant insights, especially during the revision of this manuscript when other more important experiments should be prioritized. We hope that you agree.

9. *Protein load should be quantified in Figure 4b. Currently it only refers to loading of “equal volumes” but the high amounts of protein detected by the anti-H3 antibody do not enable to clearly assess equal loading between samples. In addition, the approx size of bands detected in the western blots should be indicated in the figures.*

Response: To better assess equal loading, we have re-run the Western blot using the same protein samples but with less antibody and substrate to better show individual protein band signals (See revised Fig. 4b). A Coomassie stained gel has also been provided to show overall protein loading (See new Supplementary Fig. 9a). Finally, we have added protein size labels to the figures.

Responses to Reviewer #2:

Thank you for your input on our manuscript. We have addressed your concerns as follows:

Comments:

After showing that the synthesis of SA is compromised at 30°C, the authors excluded candidate PIFs as negative regulators of immunity at elevated temperatures. The dominant question, which heat labile component (PhyB?) regulates the activation of SA biosynthesis was not further addressed.

Response: Thank you for pointing out this timely experiment. The timing of the publication of phyB as a temperature sensor in plants did not allow us to address its role in our initial submission. However, we have since conducted disease assays using the temperature stable phyB plants, and have observed no recovery of disease resistance (See new Supplementary Fig 4b, page 7 – 8, lines 140 – 147). This result strengthens the conclusion that the phyB/PIF pathway is not involved in regulating temperature effects on disease development.

1. The roles of GRXS13 and GRX480 as positive regulators of SA signaling are not yet documented, references 30 and 31 do not prove this. These genes are just induced by SA. The role of GRX480 as an SA-induced negative regulator of JA/ET signaling has not yet been confirmed by loss of function analysis.

Response: Thank you for pointing this out. We have removed these two genes from Fig. 5 and Supplementary Table 3 (now Supplementary Table 5). We also replaced them with ZAT6 in the text as an example of a group B positive regulator of SA (see page 13, line 299).

2. Supplemental Table listing the 2844 DEGs: it is not easy for the reviewer to evaluate the quality of the RNAseq data: At least, the p-values should be indicated. Page 14: For the analysis based on differences of expression levels, which p value was chosen for the cut-off? It would be appreciated to have a list of group A genes, group B genes, etc along with the expression values.

Response: Following your suggestions, we have modified the supplementary file listing the DEGs to include q -values as well as Cluster and Group classifications (see Supplementary Dataset 1). Additionally, the description of the RNA-seq data analysis has been revised to improve clarity, as well as to reflect the new analyses that have been added (see new Supplementary Tables 1 – 4; p 12 – 14, lines 259 – 314). Specifically, the two analyses now included for DEG identification are 1) BTH vs. Mock for both temperatures and 2) 23 °C Mock vs. 30 °C Mock. In both analyses, a 4-fold change in expression (\log_2 value $\geq \pm 2$) and q -value < 0.01 were used as cut-off criteria for DEG determination.

After DEG identification, we analyzed BTH vs. Mock DEGs using both fold change differences (“cluster” analyses) and expression level differences in BTH-treated samples (“group” analyses). This enabled a more comprehensive analysis of how temperature and BTH treatments impacted the change in BTH-induced expression (mock vs. BTH) and temperature-mediated differences in gene expression at 23 °C BTH vs. 30 °C BTH.

With these further analyses, we agree that readers will find the data easier to comprehend and potentially useful for their own studies. Thank you.

Responses to Reviewer #3:

Thank you for your critical comments on our manuscript. We have addressed your concerns as follows:

Comment:

1a - *The authors demonstrated that DC3000 and DC3118 but not HrcC caused increased susceptibility at the elevated temperature that is independent of PIFs. First, the authors talked about climate change and its impact on plant immunity and agricultural yield. Yes, climate change is real and it's happening but one cannot realistically expect a dramatic 7C change in temperature that will favor bacterial growth, while plants never got the opportunity to co-evolve. I understand one can't fast forward in time and predict how the plant genome would shape up by the time there will be a 7C increase in temperature. However, one could perform experiments with an increasing magnitude of temperature to understand plant genome plasticity and figure out the minimum increase in temperature that would shift the balance and allow for compromised basal defense.*

Response: This is a good point. As we responded to Reviewer #1, our research was not designed to directly address the climate change issue. Instead, the aim of our study was to investigate the impact of a moderately elevated temperature on the *Arabidopsis-Pseudomonas syringae* pathosystem. Temperatures between 27 – 30 °C are considered “moderately elevated” for *Arabidopsis*²⁻⁴.

In the beginning of this study we had performed a temperature gradient study to assess *PR1* and *ICS1* marker gene expression at 15 °C, 25 °C, 30 °C, 35 °C and 40 °C. We found that the lowest temperature at which we observed complete loss of induction for both defense marker genes was 30 °C (see figure below). This is consistent with several published studies that examined effects of elevated temperature on *R*-gene-mediated resistant interactions using 30 °C^{5,6}. However, we could not find a logical place to insert this data, therefore decided to include this figure in the response letter for your information.

Figure: BTH-induction of SA marker gene expression at various temperatures. Three-week-old Col-0 plants were grown at 20 °C under 12:12 light:dark conditions and acclimated to test chambers at respective temperatures for 24 h prior to mock or BTH treatment. Tissue was collected 24 h after chemical treatment and used for RNA extraction. qPCR was used for gene expression analysis, with expression of *ICS1* (a) and *PR1* (b) normalized to the expression of *PP2AA3*. Letters indicate statistical significance based on a two-factor ANOVA with Tukey's HSD post hoc analysis (p – value < 0.05); samples sharing letters are not significantly different. Statistical analyses were conducted separately for each control/test temperature pair as shown by the vertical lines and use of different letters/symbols in each panel.

Comment:

1b - I'm also puzzled why the authors moved the plants for 48 hrs prior to pathogen infection unlike Cheng *et al* where they moved the plants into increased temperature conditions 15 minutes prior to the experiment? I'm wondering about a quick water loss since the plants showed hyponastic symptoms. How about the "heat shock" response?

Response: Thank you for pointing this out. The published studies of which we are aware that have examined the effect of elevated temperature on the disease phenotype have either used plants grown continuously at the elevated temperature tested⁷⁻⁹ or plants acclimated to the elevated temperature for 1 to 3 days^{5,10}. Cheng *et al.* did not conduct any disease assays at elevated temperature, but, rather, assessed various pattern-triggered immunity and effector-triggered immunity signaling outputs, such as marker gene induction and MAPK phosphorylation. We chose to acclimate plants to test chambers for 24 h (prior to chemical treatment) and 48 h (prior to pathogen treatment) to ensure the plants had ample time to adjust to the temperature change prior to pathogen infection. The longer temperature acclimation time for pathogen infection was necessary to allow for 24 h BTH treatment prior to pathogen infiltration in BTH-protection assays.

As plants were well-watered and kept at high humidity during the experiment, there was no indication of plants experiencing water loss. Furthermore, during the revision of this manuscript, we have conducted new experiments in plants that were (i) acclimated and kept at 23 °C following infection (23 °C → 23 °C), (ii) acclimated at 30 °C and shifted to 23 °C (30 °C → 23 °C), (iii) acclimated at 23 °C and shifted to 30 °C following infection (23 °C → 30 °C), or (iv) acclimated and kept at 30 °C following infection (30 °C → 30 °C). We observed that the post-infection temperature is more important than the pre-acclimation temperature. However, the magnitude of the difference in bacterial growth was greater in plants pre-acclimated to elevated temperature. These results deepen our understanding of the temperature effect and are added to the revised manuscript (new Supplementary Fig 3; page 7, lines 123 - 131).

Comment:

1c - The authors have performed global transcriptome but I did not see data corresponding to 23C and 30C transcriptome changes?

Response: As we responded to Reviewer #2, as part of the manuscript revision, we have identified DEGs based on 23 °C mock vs. 30 °C mock comparison of RNA-seq expression data using a 4-fold change (\log_2 value $\geq \pm 2$) and q -value < 0.01 as cut-off criteria for DEG determination (new Supplementary Dataset 1, page 12, lines 259 – 260). We have also conducted GO analysis of these genes, which is provided in Supplementary File 2 and summarized in Supplementary Table 1 (see also page 12, lines 262 – 265). Thank you.

Comment:

1d - What is the status of stomatal aperture? Should the authors consider using spray or dip inoculation as opposed to syringe infiltration to assay if stomatal immunity is intact?

Response: Disease assays using dip-inoculation⁷ and spray-inoculation¹⁰ have already been published, and show enhanced susceptibility at elevated temperature (28 °C and 27 °C, respectively), suggesting that stomatal defense is likely also compromised.

Future research is needed to specifically investigate the effect of elevated temperature on stomatal defense. However, such results, either positive or negative, should not affect the conclusions stated in this manuscript.

Comment:

1e - *The authors claimed that “PIFs may not play a major role in mediating elevated temperature-dependent enhancement of disease susceptibility to Pst DC3000”. However, the growth of bacteria on day 3 reached 10⁸. Could the authors use a relatively low dose of bacteria to start with? Alternatively, they can monitor the dynamic growth of pathogens by taking samples every 24 hrs.*

Response: Thank you for this good suggestion. We have repeated the *pifq* mutant experiments using a lower inoculum and sampling every 24 h, as suggested (See new Supplementary Fig. 4a, page 7, lines 135 – 139). We observe the *pifq* mutant plants to have a similar disease phenotype as wild type plants at each time point, even prior to bacterial growth saturation, strengthening the conclusion that PIFs are not involved in enhanced disease susceptibility.

Comment:

1f - *The authors used PnptII::avrPto-CyaA reporter construct to assess the translocation rate at a higher temperature. They found a three-fold increase in the translocation activity. However, they generalized the conclusion that “increased Pst DC3000 virulence at 30°C is linked to increased translocation of bacterial effector proteins”. Is this the only effector that is translocated rapidly or most of the effectors can be translocated faster? Are they more stable and/or accumulate more protein at a higher temperature? Given that MTI is induced at a higher temperature, it would be important to test delivery of ETI-triggering effectors such as avrB, avrRpm1, avrRpt2, avrPphB into the mutants that can't trigger corresponding NLR-mediated resistance at a higher temperature to investigate roles of these effectors in the MTI suppression.*

Response: This is a good point. As we responded to Reviewer #1, we have followed your suggestion and conducted effector translocation assays with three additional effector-CyaA strains, HopU1-CyaA, HopG1-CyaA and AvrPtoB-CyaA, available from published studies. Translocation of all these effectors showed a similar trend as was observed with AvrPto-CyaA (See revised Fig 1e, page 8 – 9, lines 162 – 172). This result strengthens the conclusion. To continue our focus on determining the effect of elevated temperature on a compatible (susceptible) plant-pathogen interaction, we think that an investigation of Avr proteins involved in R gene-mediated resistance is beyond the scope of this particular study. Results (either positive or negative) will not affect the conclusion with respect to the susceptible interaction we are studying. We hope that you would agree.

Comment:

2a- *The authors checked the expression patterns of ICS1 and PR1 upon infection with DC3000 (vacuum infiltration) and showed that the levels of expression and total cellular SA concentration are diminished at elevated temperatures. I would be intrigued to see such expression study performed at different temperature conditions as well as different time points since SA-dependent genes and SA accumulation are under circadian control. What if the authors use spray inoculation, ETI inducing DC3000 strain or HrcC?*

Response: Please see our response to comment 1a above regarding measuring gene expression at different temperatures.

Yes, we were aware of reports of circadian regulation of defense gene expression, and, therefore, were careful to sample tissues at the same time of the day (2 h after lights on, see p 22, line 509) for all experiments to avoid variations due to circadian regulation of defense genes. Also, our new group-based GO analysis shows that genes regulated by TFs involved in circadian regulation appear in the temperature-insensitive group B (see new Supplementary Table 4b). This information is added to the revised manuscript (page 14, lines 308 – 310).

As discussed above, temperature effects on specific *R*-gene-mediated interactions or pattern-triggered immunity have already been published or are being studied by other groups. Results (either positive or negative) will not affect the main conclusions of this study, which is focused on determining the effect of elevated temperature on the processes that are involved in a susceptible plant-pathogen interaction.

Comment:

2b - *Similar to my previous comment about infection experiment on the pif mutants, the authors should consider using low dose and/or early time points to monitor the growth of bacteria in ics1 mutant.*

Response: This is a good suggestion. We have repeated the *ics1* mutant experiments using a lower inoculum and sampling every 24 h, as suggested (See new Fig. 2c). At 2 dpi and 3 dpi, there is no difference in bacterial growth in *ics1* mutant plants at 30 °C vs. 23 °C. Bacterial growth is statistically significantly higher in *ics1* mutant plants at 30 °C vs 23 °C at 1 dpi; however, the magnitude of the difference is merely 3-fold, which ultimately did not impact disease outcome. However, we cannot exclude the possibility that an ICS1-independent factor may play some subtle role in increased susceptibility early in disease progression. A statement has been added to the revised manuscript (p 9, lines 185 – 188).

Comment:

3- *Diminished accumulation of SA at increased temperature is the most direct evidence in this manuscript. Given that SA levels are reduced and MYC genes are induced, it would be valuable to examine the growth of necrotrophic pathogens at higher temperature.*

Response: While we agree this would be an interesting experiment, we currently do not have expertise to perform this experiment. It is not easy to find a collaborating lab that has this expertise and

appropriate experimental setup, because this experiment demands the dedicated use of two precious growth chambers (for two temperatures) for several weeks, at a minimum.

Comment:

4- The authors focused on the master regulator of SA pathway, NPR1 and showed that NPR1's translocation to the nucleus is not affected. Given that SA is required for NPR1's nuclear translocation and SA is not accumulated at 30C, the question would be how NPR1 is translocated to the nucleus? Does heat alter the redox status of the cell?

Response: In these experiments, NPR1 nuclear translocation was enabled by the BTH treatment (see page 11, lines 229 and 236).

Comment:

5a- The authors performed RNA-seq experiment and identified over 2,800 genes that are BTH-regulated and 30C-dependent. They analyzed 23°C BTH vs. 30°C BTH) and (ii) fold change (BTH/mock). I wonder if the PI also analyzed 23 C and 30 C mock data to dissect the landscape of transcriptional changes that occur due to the change of temperature. The authors made four groups including BTH-induced but 30C low expression, BTH-induced 30C up expression and BTH-suppressed and 30C less suppressed. It seems that the amplitudes of 66% BTH-regulated genes is altered.

The authors have performed all the qRT-PCRs and RNA-seq using only one time point (24 hours). Based on these experiments, the authors suggest an SA-mediated bifurcation or qualitative topological behavior of these genes. The authors did not specify an analysis that led them to make this conclusion. What if the amplitude of the same set of genes may vary at different time points within a classified group? Keeping in mind most of these genes are under circadian control, it would be interesting to examine the transcript levels of a set of genes, at least the one in figure 5, with a fine time course qRT-PCR.

Response: As we responded to you earlier, as part of manuscript revision we have identified DEGs based on 23 °C mock vs. 30 °C mock comparison of RNA-seq expression data using a 4-fold change (\log_2 value $\geq \pm 2$) and q -value < 0.01 as cut-off criteria for DEG determination (new Supplementary Dataset 1). We have also conducted GO analysis of these genes, which is provided in Supplementary File 2 and summarized in Supplementary Table 1. Thank you.

Additionally, in light of your criticism we have also rephrased our conclusion to clarify that the bifurcation we observe is specific to the time point assessed (see page 13, line 289, and page 20, lines 468 – 469).

“...a clear temperature-sensitive bifurcation is revealed at the 24 h time point assessed in our study...”

Comment:

5b - The authors acclimated the plants for 24 hours prior to BTH spray. I was wondering whether the authors would obtain the same results if they moved the plants right after BTH spray to 23C and 30 C for qRT-PCR similar to what Cheng et al have done.

Response: We have conducted more detailed temperature acclimation experiments during manuscript revision (see response to your comment 1b and new Supplementary Fig. 3, p 7, lines 123 – 131). It is clear the post-infection temperature is more important than the pre-acclimation temperature. We feel that further experiments assessing gene expression would not add much to the main conclusions of this particular study. Our study was not intended to specifically follow the results in the Cheng et al. paper, which did not study actual bacterial infection, SA signaling or type III secretion *in vivo*. We hope that you agree.

Comment:

5c - *The authors performed K cluster analysis to group set of genes with similar expression at 24 h. Subsequently, they performed GO ontology and element enrichment analyses. It was puzzling to me how genes in clusters 2, 4 and 5 can be regulated by TGA factors simultaneously (based on the assumption that TGAs are BTH-induced but 30C low expression; group A blue genes) since they are categorized in group A, B and C. In theory, it is possible if TGAs are interacting with a set of positive or negative regulators that may positively and negatively influence the expression patterns. As it stands, however, the authors did not provide any genetic or biochemical evidence (totally hand waving). For instance, performing q-RT PCR on genes from cluster 2, 4 and 5 in tga2/5/6 mutant background. The same argument is true for WRKY and PIFs given that PIFs are not even involved in temperature-mediated disease susceptibility. Similarly, the authors just checked ICS1 and PR1 in camta mutant. How about testing the genes from cluster 6?*

Response: We agree that the description and presentation of RNA-seq analyses were a bit confusing in the original manuscript. To resolve this issue, we have added new, group-based GO and AME analyses (Supplementary Tables 3b and 4b), which were determined based on overall expression levels rather than fold change between the mock- and BTH-treated plants. The group-based AME analysis shows that TGA-regulated genes are predominantly in Group A, which contains genes compromised in overall expression level following BTH treatment at elevated temperature. We have also revised the description of the RNA-seq analysis within the text to improve clarity and incorporate the new analyses (see p 12 – 13, lines 259 – 291). We hope this provides sufficient clarification on this issue, and thank you for bringing it to our attention.

In regards to conducting further qPCR analyses, we feel that further experiments would not add much to the main conclusions of this study.

Comment:

5d - *The authors refuted model 1 by stating that they observed increased translocation of avrPto at 30C and hrcC growth at 30C is not increased. Given that SA accumulation is dampened at 30C, would the authors perform effector translocation assay on SA-sprayed plants at 30C with the same results?*

Response: Following your suggestion, we have conducted translocation assays at both temperatures following mock or BTH treatment, and show that BTH reduces the translocation of bacterial effectors at both temperatures, although the effect at 30 °C is somewhat variable across experiments. We have

added this new result to the revised manuscript (See new Fig 7g and Supplementary Fig 15, p 17, lines 378 – 388).

Comment:

5e - In model 3, the authors suggested that other hormones or hormonal nodes may suppress SA biosynthesis. I was wondering why the authors did not perform SA measurements or measured transcript accumulation of *ICS1*, their hallmark marker gene, in *aos*, *myc2/3/4*, *ein2* etc. mutants.

Response: Thank you for this comment. As described in the original manuscript, none of the mutants showed a reduction in disease development at elevated temperature, suggesting that they are not involved in the enhanced disease susceptibility phenotype at elevated temperature. I hope that you agree with us that further analyses, including SA measurements and gene expression, would not likely lead to biologically relevant results.

Again, we thank all three reviewers for their constructive comments and hope that the substantial new results and textual revisions have improved our manuscript.

References cited:

1. Cheng, C. *et al.* Plant immune response to pathogens differs with changing temperatures. *Nat. Commun.* **4**, 2530 (2013).
2. Lorenzo, C.D., Sanchez-Lamas, M., Antonietti, M.S. & Cerdan, P.D. Emerging hubs in plant light and temperature signaling. *Photochem. Photobiol.* **92**, 3-13 (2016).
3. Liu, J.Z., Feng, L.L., Li, J.M. & He, Z.H. Genetic and epigenetic control of plant heat responses. *Front. Plant Sci.* **6**, 21 (2015).
4. Quint, M. *et al.* Molecular and genetic control of plant thermomorphogenesis. *Nat. Plants* **2**, 9 (2016).
5. Menna, A., Nguyen, D., Guttman, D.S. & Desveaux, D. Elevated temperature differentially influences effector-triggered immunity outputs in *Arabidopsis*. *Front. Plant Sci.* **6**, 995 (2015).
6. Xiao, S.Y., Brown, S., Patrick, E., Brearley, C. & Turner, J.G. Enhanced transcription of the *Arabidopsis* disease resistance genes *RPW8.1* and *RPW8.2* via a salicylic acid-dependent amplification circuit is required for hypersensitive cell death. *Plant Cell* **15**, 33-45 (2003).
7. Wang, Y., Bao, Z., Zhu, Y. & Hua, J. Analysis of temperature modulation of plant defense against biotrophic microbes. *Mol. Plant Microbe Interact.* **22**, 498-506 (2009).
8. Carstens, M. *et al.* Increased resistance to biotrophic pathogens in the *Arabidopsis* *Constitutive Induced Resistance 1* mutant is EDS1 and PAD4-dependent and modulated by environmental temperature. *PLoS One* **9**, e109853 (2014).
9. Disch, E.M. *et al.* Membrane-associated ubiquitin ligase SAUL1 suppresses temperature- and humidity-dependent autoimmunity in *Arabidopsis*. *Mol. Plant Microbe Interact.* (2015).
10. Gangappa, S.N., Berriri, S. & Kumar, S.V. PIF4 coordinates thermosensory growth and immunity in *Arabidopsis*. *Curr. Biol.* **27**, 1 - 7 (2017).

REVIEWERS' COMMENTS:

Reviewer #1 (Remarks to the Author):

In this new version of the manuscript authors have improved the presentation of the results, highlighting the novelty and significance of their work. Authors provide additional experimental data to answer most of my previous questions and suggestions. Overall, I think authors have addressed and properly answered all my comments.

Reviewer #2 (Remarks to the Author):

The revised version of the manuscript is much clearer than the original version. My comments (phytochrome, details of the RNAseq experiment) have been addressed appropriately.

Reviewer #3 (Remarks to the Author):

The revised paper by Hout et al. reads better than the previous version and I also appreciate the authors taking their time to address most of the concerns/comments raised by all the Reviewers. This certainly strengthens authors' claims and the primary message now seems clearer. As mentioned in the previous review, there is no doubt that the data are valuable and can be useful resource for future research. The authors have performed new set of experiments as well as additional bioinformatics/functional analyses on the transcriptomics data including AME and GO analyses. These analyses clearly improve the quality of RNA-seq presentation. However, I'm not completely sold on the idea of bifurcation at 24 hr "...a clear temperature-sensitive bifurcation is revealed at the 24 h time point assessed in our study..."

What if the bifurcation might have occurred earlier or what has been observed is a branch of the whole transcriptome at this given time point. I also hope that authors would agree with me since this above mentioned statement is misleading. All what they observed is snapshot of changes in temperature/pathogen-mediated transcript differences at 24h. It's understandable that one may not be able to perform fine resolution dynamic RNA-seq due to technical reasons, however these data can be further supported by fine resolution time-course qRT-PCR. Owing to these arguments, I suggested authors performing qRT-PCR that may add additional value to their transcriptome data and refine the conclusion(s). I suggested before and I would like to emphasize that the authors should perform fine resolution time course qRT-PCR on a set of genes listed at least in Figure 5 B and C. Moreover, the authors should also consider performing qRT-PCR only at different time points throughout the entire manuscript (currently, they chose only one time-point i.e. 24h).

RESPONSE LETTER

Reviewer #1 (Remarks to the Author)

In this new version of the manuscript authors have improved the presentation of the results, highlighting the novelty and significance of their work. Authors provide additional experimental data to answer most of my previous questions and suggestions. Overall, I think authors have addressed and properly answered all my comments.

Thank you.

Reviewer #2 (Remarks to the Author)

The revised version of the manuscript is much clearer than the original version. My comments (phytochrome, details of the RNAseq experiment) have been addressed appropriately.

Thank you.

Reviewer #3 (Remarks to the Author)

The revised paper by Hout et al. reads better than the previous version and I also appreciate the authors taking their time to address most of the concerns/comments raised by all the Reviewers. This certainly strengthens authors' claims and the primary message now seems clearer. As mentioned in the previous review, there is no doubt that the data are valuable and can be useful resource for future research. The authors have performed new set of experiments as well as additional bioinformatics/functional analyses on the transcriptomics data including AME and GO analyses. These analyses clearly improve the quality of RNA-seq presentation. However, I'm not completely sold on the idea of bifurcation at 24 hr "...a clear temperature-sensitive bifurcation is revealed at the 24 h time point assessed in our study..."

What if the bifurcation might have occurred earlier or what has been observed is a branch of the whole transcriptome at this given time point. I also hope that authors would agree with me since this above mentioned statement is misleading. All what they observed is snapshot of changes in temperature/pathogen-mediated transcript differences at 24h. It's understandable that one may not able to perform fine resolution dynamic RNA-seq due to technical reasons, however these data can be further supported by fine resolution time-course qRT-PCR. Owing to these arguments, I suggested authors performing qRT-PCR that may additional value to their transcriptome dat and refine the conclusion(s). I suggested before and I would like to emphasize that the authors should perform fine resolution time course qRT-PCR on a set of genes listed at least in Figure 5 B and C. Moreover, the authors should also consider performing qRT-PCR only at different time points throughout the entire manuscript (currently, they chose only one time-point i.e. 24h).

41 In response to this concern and following the editor's guidance, we have added additional
42 clarification near the end of the discussion on page 21.

43 "We chose the 24 h time point for gene expression analysis because this is when *PR1*,
44 *PR2* and *PR5* genes are most highly expressed following BTH treatment (Lawton *et al.*,
45 1996, *Plant J*). However, it is important to note that this provides only a snap shot of the
46 dynamic transcriptional landscape, and future research should examine whether
47 temperature-sensitive genes may be induced or suppressed at earlier or later time
48 points."

49